# Epidemiology of pre-cancerous cervical lesion and risk factors among adult women in Tigray, Ethiopia

**Gerezgiher B. Abera** [1]*, **Henock G. Yebyo**[2], **Haftamu Hailekiros**[3], **Selam Niguse**[3], **Yibrah Berhe**[3], **Goitom Gigar**[4], **Tsehaye Asmelash**[5], **Gelila Goba**[6]

1 School of Nursing, College of Health Sciences, Mekelle University, Mekelle, Ethiopia, 2 School of Public Health, College of Health Sciences, Mekelle University, Mekelle, Ethiopia, 3 School of Medicine, College of Health Sciences, Mekelle University, Mekelle, Ethiopia, 4 Tigray Regional Health Bureau, Mekelle, Ethiopia, 5 College of Health Sciences, Aksum University, Axum, Ethiopia, 6 Department of Obstetrics and Gynecology, University of Illinois at Chicago, Chicago, Illinois, United States of America

* gbamsc2002@gmail.com

**Data Availability Statement:** All data that are used to analyze this paper are within the paper and its Supporting Information files.

## Abstract

### Background

Cervical cancer is a preventable disease if treated early, but remains the second leading cause of cancer-related mortality among women in low and middle-income countries. Data on epidemiology and risk factors in these settings are scarce. This study aimed to assess the prevalence of pre-cancerous cervical lesions and risk factors in Tigray region, Ethiopia.

### Methods

A community-based, cross-sectional study was used and 900 participants were 30 recruited using multistage sampling and finally data from 883 were collected using an interviewer administered questionnaire and screening with visual inspection with ascetic acid. Data were collected using an interviewer administered questionnaire and screening with visual inspection with acetic acid from March 2016 to June 2017. Multinomial logistic regression analysis was conducted to estimate predictors.

### Results

Seventy-nine (8.95%) women were positive for pre-cancer lesion and 35 (3.96%) were suspicious for cervical cancer. We used relative risk ratio (rrr) to estimate the strength of association. Divorced or widowed women had 2.5 and 4.7 times more risk of being positive and suspicious respectively, compared to single women (rrr = 2.5, 95% CI [1.13, 5.52]); (rrr = 4.69, 95% CI [1.00, 21.84]). The risk of having a suspicious result was 68% lower for women with primary education compared to those with no formal education (rrr = 0.32, 95% CI [1.00, 21.84]). History of sexually transmitted infection was associated with positive pre cancer lesion (rrr = 1.91, 95% CI [1.11, 3.27]) whereas, being farmer (rrr = 4.83, 95% CI [1.44, 16.13]), merchant (rrr = 4.85, 95% CI [1.52, 15.46]), bleeding between periods (rrr = 3.26,

**Funding:** This project is funded by Mekelle University, Mekelle, Ethiopia and Tiray Regiona Health Bureau with the registration number of CRPO/CHS/Ext001/08/2015. "The author(s) received no any funding for this paperwork or manuscript publication from any source." "The Mekelle University and Regional Health Bureau had no role in study design, data collection and analysis, decision to publish, or preparation of the manuscript."

**Competing interests:** The authors have no competing interests to declare

**Abbreviations:** aRRR, Adjusted Relative Risk Ratio; cRRR, Crude Relative Risk Ratio; FMoH, Ethiopia's federal ministry of health; HEWs, Health Extension Workers; HIC, High Income Countries; HIV, Human Immunodeficiency Virus; HPV, Human Papilloma Virus; LMICs, low-and-middle income countries; RRR, Relative Risk Ration; SCJ, Squamous-Columnar Junction; SD, Standard Deviation; STATA, STATistical Analysis; STI, Sexually Transmitted Infection; TRHB, Tigray Regional Health Bureau; VIA, Visual Inspection With Acetic acid; WDA, Women Development Army.

95% CI [1.32, 8.04]) and pelvic or back pain (rrr = 2.79, 95% CI [1.18, 6.58]) were associated with suspicious for cancer.

## Conclusion

About 8.9% and 3.96% of the women were positive for pre-cancerous cervical lesion and suspicious for cancer, respectively. The prevalence of pre-cancerous cervical lesion is high as compared to other regional prevalence in the country. Marital status, education, sexually transmitted infection, bleeding, and pelvic pain were risk factors of pre-cancerous cervical lesion'. This finding implies that the sexual exposure, having no permanent husband and being not educated attributes to the high prevalence of pre-cancerous cervical lesion and may aggravate the transmission of HPV."

## Introduction

Cervical cancer is a public health problem worldwide [1], primarily in low and middle-income countries, though data are scarce [2, 3]. The disease is fatal, but preventable if treated early [4]. The World Health Organization (WHO) indicates that cervical cancer kills an estimated 275,000 women every year and 500,000 new cases reported worldwide [1]. Cervical cancer is the second largest killer among cancer-related death of women in low-and-middle income countries (LMICs) [4]. Reports from different countries showed that the incidence of cervical cancer is high in developing countries and sexually transmitted infections (STIs), hormonal influences, genetics and participant factors (risk taking behavior, substance abuse, alcohol use) are risk factors for cervical cancer [5, 6]. In Sub-Saharan Africa, an estimated 57,000 cases of cervical cancer occurred in 2000, comprising 22.2% of all female cancers [7].

A study from India revealed that late-stages reporting of disease was common with a peak age of cervical cancer incidence from 55 to 59 years and early marriage, multiple sexual partners, multiple pregnancies, oral contraceptives, and lack of awareness were significant risk factors [8]. A study in Uganda showed that 99% of women had heard about cervical cancer and 63% believed that family planning was a cause of cervical cancer as well as 85% recognizing inter-menstrual bleeding as a symptom of cervical cancer [9]. Other studies have also revealed the burden and public health importance of cervical cancer in LMICs [10–13].

In Ethiopia, cervical cancer is the most common cause of cancer death among female cancers with an overall mortality of 70% [14]. A study in Ethiopia showed that the mean age at diagnosis with cervical cancer was 48 years [15]. In a community based cross-sectional survey conducted in Gondar, northwest Ethiopia (2010), only 31% of respondents were knowledgeable about risk factors, symptoms, treatment options, prevention and early detection measures for cervical cancer [16]. Studies from Addis Ababa, Adama and Jimma showed that women 40–49 years of age, having a history of STIs, having two or more lifetime sexual partners and contraceptive use were risk factors of the cervical pre-cancerous lesion [17–19]. Another institution based studies in Tigray revealed a prevalence of pre-cancerous cervical lesion of 6.7% and that STIs are predictors of the pre-cancerous cervical lesions [20, 21].

The Ethiopian Ministry of Health officially launched guideline for cervical cancer prevention and control in 2005, describes a "See and Treat approach" using visual inspection with acetic acid (VIA) screening and cryotherapy treatment for the reason of feasibility and affordability [22].

Although Ethiopia's federal ministry of health (FMoH) has given priority for cervical cancer prevention and control, there is a scarcity of data related to pre-cancerous cervical lesions. Existing guidelines call for community based pre-cancerous cervical lesions epidemiology and risk factors [22]. Accordingly, this community-based research determines the prevalence and factors associated with pre-cancerous cervical lesions among adult women in Tigray, Ethiopia.

## Materials and methods

### Ethical consideration

The study protocol was evaluated and approved by the Research Ethics Review Committee of College of Health Sciences, Mekelle University Research and Community Service Committee (Registration number–ERC 0597/2015). Official cooperation letters were obtained from Mekelle University, Tigray Regional Health Bureau, and zonal health offices. Moreover, prior conducting the study, the purpose and objective of the study were described to the study participants and a written informed consent was obtained. The consent involves permission to disseminate the findings of the study through scientific workshop and publish in reputable journals. The study participants were told that they have full right to discontinue the study. Confidentiality and any special data security requirements were maintained and assured. The results of the screening that have a direct benefit to the health of the study participants were informed to physicians working in the nearby health facility.

### Study setting and design

We conducted a community-based, cross-sectional study in Tigray, Northern Ethiopia. Data were collected from March 2016 to June 2017. The study was conducted in Tigray region, which has seven zones, 34 rural and 18 urban districts ('Woredas' in local name) and 763 sub-districts ('Kebele/Tabia' in local name). In 2017, the Tigray Region had one specialized hospital, 14 general hospitals, 22 primary hospitals, 214 health centers, and 712 health posts as well as more than 500 private health facilities.

### Study population

**Study participants.**   All adult women living in Tigray region for about six months or more were considered as a source population, and those who were voluntary and provided informed consent were study participants.

**Eligibility criteria.**   All adult women, who were sexually active and voluntarily consented to participate were included in the study. Those with history of hysterectomy were excluded from the study.

### Sample size and sampling technique

The sample size was calculated with the aim of estimating a precise population prevalence of pre-cancerous cervical lesions in the Tigray Region. As there is no baseline study of prevalence of pre-cancerious cervical lesion, we assumed a 0.5 probability of the disease, 0.05 probability of alpha testing, 80% power and design effect of 2 accounting for two-stage sampling. Previous studies, the response rate ranges from 77.1% to 98.2%, which implies a 1.8% to 2.9.9% contingency [23, 24]. We added 10% to the calculated sample size to account for non-responses, yielding a sample size of 846, which we rounded to 900 for convenience.

A multistage sampling was used to select the study subjects. Considering 40% of the 52 districts in the region, 21 of the districts which were distributed proportionally to zones were selected randomly. Since there are 315 sub-districts in the selected districts, 20% of the sub-

districts (a total of 63 sub-districts) were selected randomly. Study participants were allocated proportionally to sub-districts based on the number of participants fulfilling the eligibility criteria in each sites. This was taken from the health extension workers registration book.

**Recruitment process.** A community mobilization exercise preceded the screening. This effort entailed training of Health Extension Workers (HEWs) from selected sub-districts. HEWs in turn oriented women in their communities. They also mobilized Women Development Army (WDA) leaders and members as well as sub-district leaders by providing brief orientations about cervical cancer as well as upcoming screening dates and locations. Grassroots mobilization was complimented by promotion on regional radio stations by the Tigray Regional Health Bureau (TRHB).

Following community mobilization, participants were invited for screening to a total of 17 primary and general hospitals in the sub districts which have cervical cancer screening cencers using visual inspection with acetic acid (VIA). As per the estimated number of expected participants, a key value of 4 was calculated. Every forth woman arriving at the hospitals following grassroots mobilization were selected and invited to participate in the study. Finally, those who fulfill the eligibility criteria were recruited as a study subject.

## Data collection procedure

Women providing consent were interviewed using an interviewer administered questionnaire to obtain socio-demographic and risk-factors information, including health related behaviors and lifestyle, knowledge and attitude of cervical cancer screening, and clinical attributes.

Clinical testing was conducted by certified nurses trained by the TRHB according to the national cervical cancer prevention and control guidelines [25]. A VIA screening protocol was developed based on the national guideline. The protocol included private examination area, examination table, trained health professionals, adequate light source, sterile vaginal speculum, new examination or surgical gloves, large cotton swabs, diluted (3–5%) acetic acid, small bowl, containers with 0.5% chlorine solution, plastic bucket with a plastic bag and the steps as quality assurance system to maximize accuracy.

Pre-cancerous cervical lesion was screened using vaginal speculum examination during which providers applied diluted acetic acid to the cervix. The provider was supported by the guideline protocol of VIA screening and supervised by gynecologists available in the institutions and assigned supervisors as appropriate to assure the quality of data. Visual confirmation of color change by the naked eye was used to determine the presence or absence of lesion. The test was considered positive if a sharp, distinct, well-defined, dense (opaque/dull or oyster) Aceto-white area was present with raised margins touching the Squamo-columnar junction (SCJ). We assumed the test as negative if no or faint Aceto-white lesions and or smooth surface was present. The test was taken as suspicious for cervical cancer where clinically visible ulcerative, cauliflower like growth or ulcer, oozing, and/or bleeding on touch were observed.

The history of sexually transmitted infection (STI) was determined if any one of the sign and symptoms (discharge, offensive secretion, itching, dysuria, lower abdominal pain, and fever) were present. The variable 'history of cervical cancer screening' was not included in our study, since most of our sites did not have the screening modality previously and we may get no participants with previous history of cervical cancer screening.

## Data analysis

Data were analyzed using statistical analysis (STATA) version14. Data entry and clearance was done using descriptive analysis to check if any missing and inappropriate categories. Descriptive statistics were presented in texts and tables. The dependent variable was pre-cancerous

cervical lesion'status, in which the screening result had three possible outcomes: Negative, Positive and Suspicious for VIA. A multinomial regression model was used to check the influence of socio-demographic, risk factors and reproductive health characteristics on pre-cancerous cervical lesion 'status. Unadjusted multinomial regression was used to examine the separate bivariate relationship between each independent variables and the dependent variable. The variables with p-value < 0.05 in the unadjusted multinomial regression was considered as significant and used in the multivariable multinomial logistic regression. Relative risk ratios (rrr) were used to assess the multivariate association of the predictor variables with the dependent variable.

## Results

### Socio-demographic characteristics of the study participants

A total of 883 questionnaires were collected with complete information, equating to a response rate of 98.1%. More than half of the participants 496 (56.2%) came from urban districts. Mean age of participants was 35.69 years (standard deviation (SD) ±8.45, range 16–62). Among all respondents, 532 (60.2%) were married, 723 (81.9%) were Orthodox Christian, 286 (32.4%) had no formal education, 282 (31.9%) had no formal work, and 717 (81.2%) had zero to four children. Median income was Ethiopian Birr 1,500 and 515 (58.3%) women earned between Ethiopian Birr 0 and 2,000 per month [Table 1].

### Risk factors of cervical cancer

A total of 782 (88.6%) had visible SCJ. Among all the participants, 79 (8.9%) were positive for VIA, 35 (3.96%) were suspicious for cancer, and 769 (87.1%) were negative for pre-cancer lesions. Participants with positive result for VIA and suspicious for cervical cancer were linked to related medical centers for further investigation and management. Most VIA positive participants were treated with Cryotherapy in the same site, while those who are not cryothrerapy eligible and suspicious results were refered to centers having loop electro-excision procedure (LEEP) and pathology for further investigation and management. A total of 776 (87.9%) reported that their age at first sexual contact was less than 20 years. Of the total, 344 (39.0%) respondents had more than two sexual partners in their lifetime and 199 (22.5%) experienced STIs. Thirty-two (3.6%) were smokers and one-third of respondents consumed alcohol 255 (28.9%). Corticosteroids were used in 97 (11.0%) women and 199 (22.5%) where current contraceptive users. Seventy (7.9%) had a family history of CC. Out of these women, nine (12.9%) were VIA-positive. The variable HIV status was excluded from analysis, since it was filled incomplete, because some sites has no HIV testing clinic and filled as unknown, which could not be logically true to treat those sites with sites that have HIV clinic [Table 2].

### Signs and symptoms of cervical cancer

From the study participants, 183 (20.7%) women experienced bleeding between menstruation, 94 (10.6%) experienced heavy menstruation, 176 (19.9%) experienced contact bleeding and 152 (17.2%) women experienced pain during sex. Pelvic or back pain was reported by 267 (30.2%) women [Table 3].

### Factors associated with pre-cancerous cervical lesion

In the multinomial analysis "Negative result of the dependent variable was used as a base reference against the positive and suspicious results; hence the relative risk ratio (rrr) is relative to those who have negative results.

**Table 1. Distribution of socio-demographic characteristics by pre-cancerous cervical lesion status among women of reproductive age in Tigray, (n = 883).**

| Socio-demographic Variables | Groups | VIA results | | | Total |
|---|---|---|---|---|---|
| | | Negative | Positive | Suspicious | |
| | | N (%) | N (%) | N (%) | N (%) |
| Residence | Rural | 345(89.1) | 28(7.2) | 14(3.6) | 387(43.8) |
| | Urban | 424(85.5) | 51(10.3) | 21(4.2) | 496(56.2) |
| Age (yr) | 15–30 | 257(86.5) | 30(10.1) | 10(3.4) | 297 (33.6) |
| | 31–45 | 416(87.6) | 44(9.3) | 15(3.2) | 475 (53.8) |
| | > = 46 | 96(86.5) | 5(4.5) | 10(9.0) | 111 (12.6) |
| Marital status | Single | 208(92.0) | 14(6.2) | 4(1.8) | 226(25.6) |
| | Married | 472(88.7) | 42(7.9) | 18(3.4) | 532(60.2) |
| | Divorced/Others | 89(71.2) | 23(18.4) | 13(10.4) | 125(14.2) |
| Religion | Orthodox | 627(86.7) | 64(8.9) | 32(4.4) | 723(81.9) |
| | Muslim and others | 142(88.8) | 15(9.4) | 3(1.9) | 160(18.1) |
| Educational status | No formal education | 240(83.9) | 25(8.7) | 21(7.3) | 286(32.4) |
| | Primary education | 325(89.0) | 30(8.2) | 10(2.7) | 365(41.3) |
| | Secondary education | 116(85.3) | 17(12.5) | 3(2.2) | 136(15.4) |
| | Tertiary education | 88(91.7) | 7(7.3) | 1(1.0) | 96(10.9) |
| Occupation | Civil servant | 209(93.3) | 11(4.9) | 4(1.8) | 224(25.4) |
| | Farmer | 141(81.5) | 21(12.1) | 11(6.4) | 173(19.6) |
| | Merchant | 172(84.3) | 18(8.8) | 14(6.9) | 204(23.1) |
| | No formal work | 247(87.6) | 29(10.3) | 6(2.1) | 282(31.9) |
| Monthly income | 0–2000 | 439(85.2) | 51(9.9) | 25(4.9) | 515(58.3) |
| | 2001–4000 | 158(86.3) | 16(8.7) | 9(4.9) | 183(20.7) |
| | 4001–6000 | 105(92.1) | 8(7.0) | 1(0.9) | 114(12.9) |
| | 6001–8000 | 16(88.9) | 2(11.1) | 0(0.0) | 18(2.0) |
| | 8001–10000 | 51(96.2) | 2(3.8) | 0(0.0) | 53(6.0) |
| Parity | 0–4 children | 628(87.6) | 67(9.3) | 22(3.1) | 717(81.2) |
| | 5–12 children | 141(84.9) | 12(7.2) | 13(7.8) | 166(18.79) |

Multinomial bivariate analysis revealed that age, marital and education status, occupation, symptoms of cervical cancer, parity, STI history of partner, and lower abdominal pain were significantly associated with positive or suspicious results (p-value < 0.05). Other variables were not significantly associated with pre-cancerous cervical lesion'. The multivariate logistic regression analysis showed that marital and educational status, occupation, history of STIs, bleeding between menstruation, and pelvic or back pain were significantly associated with the presence of pre-cancerous cervical lesion'. Divorced or widowed women were 2.5 and 4.7 times more risk of being VIA-positive and suspicious respectively than single women (rrr = 2.50, 95% confidence interval (CI) [1.13, 5.52]); (rrr = 4.69, 95% CI [1.00, 21.84]). The risk of having lesions suspicious of cancer was 68% lower for women with primary education than those with no formal education (rrr = 0.32, 95% CI [1.00, 21.84]). The risk of being suspicious for cancer was 4.8 times higher among farmers (rrr = 4.83, 95% CI [1.44, 16.13]) and merchants (rrr = 4.85, 95% CI [1.52, 15.46]) than respondents with no formal work.

The risk of being VIA-positive was 90% higher for women with a history of STIs (rrr = 1.91, 95% CI [1.11, 3.27]) than their counter parts. The risk of being suspicious for cancer was 3.26 times higher for women who experienced bleeding between periods (rrr = 3.26, 95% CI [1.32, 8.04]) than their counter parts. Women with a history of pelvic or back pain were 2.7 times more likely to be suspicious for cancer (rrr = 2.79, 95% CI [1.18, 6.58]) [Table 4] than their counter parts.

**Table 2. Distribution of risk factors by pre-cancerous cervical lesion status among women of reproductive age in Tigray, (n = 883).**

| Risk factors of cervical cancer | | VIA results | | | Total |
|---|---|---|---|---|---|
| | | Negative | Positive | Suspicious | N (%) |
| | | N (%) | N (%) | N (%) | |
| Age at first sexual contact | ≤ 20 years age | 675(87.0) | 66(8.5) | 35(4.5) | 776(87.9) |
| | > 20 years age | 94(87.9) | 13(12.1) | 0(0.0) | 107(12.1) |
| Number of Sexual Partner | Only one | 461(88.3) | 45(8.6) | 16(3.1) | 522(59.1) |
| | Two or more | 293(85.2) | 33(9.6) | 18(5.2) | 344(39.0) |
| | No or unspecified | 15(88.2) | 1(5.9) | 1(5.9) | 17(1.9) |
| History of STI | Yes | 156(78.4) | 32(16.1) | 11(5.5) | 199(22.5) |
| | No | 613(89.6) | 47(6.9) | 24(3.5) | 684(77.5) |
| Active smoking | Yes | 29(90.6) | 2(6.2) | 1(3.1) | 32(3.6) |
| | No | 740(87.0) | 77(9.0) | 34(4.0) | 851(96.4) |
| Alcohol consumption | Yes | 232(91.0) | 18(7.1) | 5(2.0) | 255(28.9) |
| | No | 537(85.5) | 61(9.7) | 30(4.8) | 628(71.1) |
| Chronic corticosteroids use | Yes | 86(88.7) | 8(8.2) | 3(3.1) | 97(11.0) |
| | No | 683(86.9) | 71(9.0) | 32(4.1) | 786(89.0) |
| Current contraceptive use | Yes | 169(84.9) | 21(10.6) | 9(4.5) | 199(22.5) |
| | No | 600(87.7) | 58(8.5) | 26(3.8) | 684(77.5) |
| Family history of cervical cancer | Yes | 60(85.7) | 9(12.9) | 1(1.4) | 70(7.9) |
| | No | 709(87.2) | 70(8.6) | 34(4.2) | 813(92.1) |

NB: STI–sexually transmitted infection

## Discussion

Ethiopia's Federal Ministry of Health has prioritized cervical cancer prevention and treatment [25]; however, there is a scarcity of data related to the prevalence of pre-cancerous cervical lesions. This study explored the prevalence of pre-cancerous cervical lesions in Tigray, Northern Ethiopia.

**Table 3. Distribution of signs and symptoms by pre-cancerous cervical lesion status among women of reproductive age in Tigray, (n = 883).**

| Sign and symptoms and cervical cancer status | | VIA results | | | N (%) |
|---|---|---|---|---|---|
| | | Negative | Positive | Suspicious | |
| | | N (%) | N (%) | N (%) | |
| Bleeding between menstruation | Yes | 149(81.4) | 16(8.7) | 18(9.8) | 183(20.7) |
| | No | 620(88.6) | 63(9.0) | 17(2.4) | 700(79.3) |
| Heavy menstruation | Yes | 70(74.5) | 12(12.8) | 12(12.8) | 94(10.6) |
| | No | 699(88.6) | 67(8.5) | 23(2.9) | 789(89.4) |
| Bleeding after menopause | Yes | 78(77.2) | 14(13.9) | 9(8.9) | 101(11.4) |
| | No | 691(88.4) | 65(8.3) | 26(3.3) | 782(88.6) |
| Contact bleeding | Yes | 143(81.2) | 19(10.8) | 14(8.0) | 176(19.9) |
| | No | 626(88.5) | 60(8.5) | 21(3.0) | 707(80.1) |
| Pain during sex | Yes | 120(78.9) | 23(15.1) | 9(5.9) | 152(17.2) |
| | No | 649(88.8) | 56(7.7) | 26(3.6) | 731(82.8) |
| Pelvic or back pain | Yes | 213(79.8) | 33(12.4) | 21(7.9) | 267(30.2) |
| | No | 556(90.3) | 46(7.5) | 14(2.3) | 616(69.8) |
| Unusual vaginal discharge | Yes | 239(79.9) | 37(12.4) | 23(7.7) | 299(33.9) |
| | No | 530(90.8) | 42(7.2) | 12(2.1) | 584(66.1) |

**Table 4. Bivariate and multivariable analysis of independent variables with pre-cancerous cervical lesion status among reproductive age women in Tigray, (n = 883).**

| Variables | | Negative | VIA results | | | | | |
|---|---|---|---|---|---|---|---|---|
| | | | Positive | | | Suspicious for cancer | | |
| | | N(%) | N(%) | cRRR [95% CI] | aRRR [95% C.I.) | N(%) | cRRR [95% CI] | aRRR [95% C.I.) |
| Age | 15–30 | 257(86.5) | 30(10.1) | 1 | 1 | 14(3.6) | 1 | 1 |
| | 31–45 | 416(87.6) | 44(9.3) | 0.91 (0.56, 1.48) | 0.50(0.17, 1.45) | 21(4.2) | 0.93(0.410, 2.09) | 2.34(0.74, 7.36) |
| | > = 46 | 96(86.5) | 5(4.5) | 0.45(0.17, 1.18) | 0.91(0.53, 1.55) | 10(3.4) | 2.68(1.08, 6.63) * | 0.58(0.22, 1.55) |
| Marital status | Single | 208(92.0) | 14(6.2) | 1 | 1 | 4(1.8) | 1 | 1 |
| | Married | 472(88.7) | 42(7.9) | 1.32(0.71, 2.47) | 0.94(0.46, 1.91) | 18(3.4) | 1.98(0.66, 5.93) | 2.15(0.51, 8.97) |
| | Divorced/widowed | 89(71.2) | 23(18.4) | 3.84(1.89, 7.80)*** | 2.50(1.13, 5.52)* | 13(10.4) | 7.59(2.41, 23.93)*** | 4.69(1.00, 21.84)* |
| Educational status | No formal education | 240(83.9) | 25(8.7) | 1 | 1 | 21(7.3) | 1 | 1 |
| | Primary Education | 325(89.0) | 30(8.2) | 0.89(0.51, 1.55) | 0.78(0.41, 1.48) | 10(2.7) | 0.35(0.17, 0.76) *** | 0.32(0.11, 0.91)* |
| | Secondary Education | 116(85.3) | 17(12.5) | 1.41(0.73, 2.71) | 1.15(0.54, 2.43) | 3(2.2) | 0.29(0.09, 1.01) | 0.25(0.05, 1.09) |
| | Tertiary Education | 88(91.7) | 7(7.3) | 0.76(0.32, 1.83) | 0.84(0.29, 2.39) | 1(1.0) | 0.13(0.02, 0.98)* | 0.12(0.01, 1.41) |
| Occupation | Civil Servant | 209(93.3) | 11(4.9) | 0.45(0.22, 0.92)* | 0.78(0.41, 1.48) | 4(1.8) | 0.78(0.22, 2.82) | 2.67(0.48, 14.68) |
| | Farmer | 141(81.5) | 21(12.1) | 1.26(0.69, 2.30) | 1.15(0.54, 2.43) | 11(6.4) | 3.21(1.16, 8.87)* | 4.83(1.44, 16.13)* |
| | Merchant | 172(84.3) | 18(8.8) | 0.89(0.47, 1.65) | 0.84(0.29, 2.39) | 14(6.9) | 3.35(1.26, 8.89)* | 4.85(1.52, 15.46)** |
| | No formal work | 247(87.6) | 29(10.3) | 1 | 1 | 6(2.1) | 1 | 1 |
| Parity | 0–4 children | 628(87.6) | 67(9.3) | 1 | 1 | 22(3.1) | 1 | |
| | 5–12 children | 141(84.9) | 12(7.2) | 0.79(0.42, 1.51) | 0.85(0.40, 1.79) | 13(7.8) | 2.63(1.29, 5.35)** | 1.62(0.61, 4.25) |
| History of STI | Yes | 156(78.4) | 32(16.1) | 2.67(1.65, 4.33)*** | 1.91(1.11, 3.27)* | 11(5.5) | 1.80(0.86, 3.75) | 0.99(0.40, 2.42) |
| | No | 613(89.6) | 47(6.9) | 1 | 1 | 24(3.5) | 1 | 1 |
| Blood between period | Yes | 149(81.4) | 16(8.7) | 1.05(0.59, 1.88) | 0.71(0.35, 1.45) | 18(9.8) | 4.40(2.21, 8.75)*** | 3.26(1.32, 8.04)* |
| | No | 620(88.6) | 63(9.0) | 1 | 1 | 17(2.4) | 1 | 1 |
| Heavy menstruation | Yes | 70(74.5) | 12(12.8) | 1.78(0.92, 3.46) | 1.23(0.55, 2.74) | 12(12.8) | 5.20(2.48, 10.9)*** | 2.45(0.92, 6.52) |
| | No | 699(88.6) | 67(8.5) | 1 | 1 | 23(2.9) | 1 | 1 |
| Bleeding after menopause | Yes | 78(77.2) | 14(13.9) | 1.90 (1.02, 3.55)* | 1.25(0.57, 2.71) | 9(8.9) | 3.06(1.38, 6.77) ** | 1.27(0.40, 3.99) |
| | No | 691(88.4) | 65(8.3) | 1 | 1 | 26(3.3) | 1 | 1 |
| Contact bleeding | Yes | 143(81.2) | 19(10.8) | 1.38(0.80, 2.39) | 0.85(.42, 1.69) | 14(8.0) | 2.91(1.44, 5.87)** | 0.81(0.30, 2.19) |
| | No | 626(88.5) | 60(8.5) | 1 | 1 | 21(3.0) | 1 | 1 |
| Pain during sex | Yes | 120(78.9) | 23(15.1) | 2.22(1.31, 3.74)** | 1.39(0.70, 2.74) | 9(5.9) | 1.87(0.85, 4.09) | 0.69(0.23, 2.01) |
| | No | 649(88.8) | 56(7.7) | 1 | 1 | 26(3.6) | 1 | 1 |
| Pelvic or back pain | Yes | 213(79.8) | 33(12.4) | 1.87(1.16, 3.00)* | 1.60(0.90, 2.85) | 21(7.9) | 3.91(1.95, 7.84)*** | 2.79(1.18, 6.58)* |
| | No | 556(90.3) | 46(7.5) | 1 | 1 | 14(2.3) | 1 | 1 |
| Unusual vagina discharge | Yes | 239(79.9) | 37(12.4) | 1.95(1.22, 3.11)** | 1.39(0.78, 2.49) | 23(7.7) | 4.26 (2.08, 8.68)*** | 2.22(0.91, 5.38) |
| | No | 530(90.8) | 42(7.2) | 1 | 1 | 12(2.1) | 1 | 1 |

NB: P—value < 0.001 = ***,

0.001–0. 009 = ** and

0. 010–0. 05 = * and rrr = relative risk ratio, cRRR–crude relative risk ratio, aRRR—Adjusted relative risk ratio, C.I.—Confidence Interval

Our study showed that 79 (8.9%) respondents were VIA-positive and 35 (3.96%) were suspicious for cervical cancer. The overall prevalence of VIA-positive was higher in this study than a study at Alameda Textile Factory in Tigray, which showed a VIA-positive rate of 6.7% [20], which may be the fact that most the workers of the textile have similar lifestyle and are living in one town of Tigray. A study at a Family Guidance Association (FGA) Clinic in Jimma, southern Ethiopia, showed that 12.9% of women were VIA-positive and 1.2% of women were suspicious for cancer [19]. Those who attend the Family Guidance Association are those who feel that they need cervical screening which could have the probability of being VIA-positive.

The long time service of cervical screening at family guidances prevents the pre-cancerous lesion from progression to suspicious for cancer. Studies in Swaziland, Cameroon, and Jakarta, Indonesia, showed VIA-positive rates of 15%, 3.33%, and 4.7%, respectively [10, 11, 12]. Though most are similar, the difference might be because of the skill difference of the care providers, and type of screening used.

After adjusting for other variables, marital and educational status, occupation, history of STIs, bleeding between period, and pelvic or back pain were statistically significant risk factors for VIA-positive or suspicious for cancer.

Divorced or widowed women had a risk of being VIA-positive (p = 0.023) and being suspicious (p = 0.049) for cervical cancer than single women. A study done in Dar Es Salaam, Tanzania, showed that divorced women were more likely to be VIA positive [13], which is similar to this study. A study in Jakarta, Indonesia, revealed that the overall VIA positive rate was 4.7%; of which, 8.3% were married more than one times in their life [12]. This implies that women who have started sexual practice and have divorced, widowed or separated and or without having a permanent husband are likely to be exposed to multiple sexual partner; which increases the risk of HPV infection. HPV infection increases the risk of cervical cancer development. Increasing outreach and screening intensity to divorced, separated, and widowed women could be a simple approach to streamlining the identification and treatment of women at risk for developing cervical cancer.

Women with primary education were approximately 70% less likely to be suspicious for cervical cancer (p = 0.034) than women with no education; and education was not significantly correlated with VIA positivity. What's more, higher education levels, including secondary and tertiary, were not significantly associated with VIA-positive or suspicious. Similar to our study, a research findings from Dar Es Salaam, showed that women with low education had odds 4.3 higher of being VIA positive [13]. Studies in Gondar, Ethiopia, and India also showed that lack of awareness on cervical cancer was a risk factor for cervical cancer [7, 16]. This might be that increasing awareness including educational level could influence individuals to prevent themselves from any risk factors of cervical cancer. Having poor knowledge on cervical cancer attributes at least the delayance of screening for cervical cancer, which gives time for the advancement of the pre-cancerous lesion to cancer.

In this study, women who are farmers and merchants in occupation had odds of 4.8 times higher of being suspicious for cervical cancer. Farmers may have less awareness of cervical cancer and lack of ability to participate in the early screening of cervical cancer. Though merchants are believed to be rich individuals, and have capacity to make health, they could also have the opportunity to visit different areas and stay out of houses. This may expose them to heightened risk due to increased chance of sexual practices that expose them to HPV.

Respondents with a history of STI had 1.91 times the risk of being VIA-positive (p = 0.018), and history of STIs were not significantly associated with suspicious results. This was similar to studies done in Addis Ababa, Adama and Alameda Textile Factory, which revealed that the history of STI was a risk factor for pre-cancerous cervical lesion [17, 18, 20]. This could be that women with STIs are more prone to aquire HPV infection, which is by far the dominant pathway for cervical cancer to initiate.

Bleeding between period and pelvic or back pain were significantly associated with suspicious cancer, which were 3.26 (p = 0.010) and 2.79 (p = 0.019) times higher risk respectively than their counter parts. This finding was similar to a study done in Uganda revealed that the participants recognized inter-menstrual bleeding as a symptom of cervical cancer [9]. This might be mainly related to suspicious for cervical cancer; as the lesion advanced, the common sign and symptom are vaginal bleeding and pain.

In this study, 8.9% of 46 years or older women were found suspicious for cervical cancer compared to approximately 9.5% of women less than 46 years of age. Age was not statistically significant. Unlike this study, a study in India revealed that women 55 to 59 years of age had the highest risk of cervical cancer [8], this was also observed in one of the studies in Ethiopian [15]. This may implies that Ethiopian women may have earlier sexual initiation than Indian women that delaying introduction and progression of the disease to pre-cancerous cervical lesion. Indian is also more developed than Ethiopia, which could has advanced screening modality and helps them prevent the development of the lesion to suspicious for cancer.

In this study, the presence of pre-cancerous cervical lesion was not associated with current contraceptive use or age at first sexual contact. Unlike to this study, the age at first sexual contact was significant risk factor for pre-cancerous cervical lesion'in a study done in Jimma [19]. Unlike to this study, a research findings done in Jakarta, Addis Ababa and Adama, Uganda, and India showed that contraceptive use was a risk factor for pre-cancerous cervical lesions [9, 8, 12, 17, 18]. Logically, long term contraceptive uses might be the risk factor for cervical cance. This could be because of the long time sexual exposure without mechanical prevention could increase the chance of HPV acquiring, which is a risk factor for cancer. Hence, it might be because of that there are no more long time contraceptive users in the study area, which is evidenced by having high fertility rate. This might be preventive to pre-cancerous cervical lesion.

## Strength and limitation of the study

The primary data of pre-cancerous cervical lesion was ascertained using active VIA screening, which minimizes recall bias. Other pre-cancerous cervical lesion screening are better identify the lesion than VIA, though the methods are not available in the study sites and are expensive than VIA.

This study used cross-sectional design for ease of screening implementation, though a case control study design could better estimate the association of independent variables with the outcome variable.

This study did not include variables that have probability of being risk factors like HIV status and history of cervical cancer screening because of the absence of the services in some sites. Though the variable HIV status is common risk factor for cervical cancer in other studies, it had been removed from analysis because of some sites have no HIV testing clinics and questionnaire form those areas were filled as unknown.

## Conclusion

In this study, 8.9% and 3.96% of the women were positive and suspicious for pre-cancerous cervical lesion respectively. The prevalence of pre-cancerous cervical lesion is high as compared to other regional prevalence in the country. Marital status, educational status, occupation, history of an STI, bleeding between period, and pelvic or back pain were risk factors for pre-cancerous cervical lesion. This finding implies that the sexual exposure, having no permanent husband and being not educated attributes to the high prevalence of pre-cancerous cervical lesion and may aggravate the transmission of HPV.

## Recommendation

We recommend that cervical cancer prevention is better to be strengthened focusing on preventable risk factors, including exposure to sexually transmitted infections and specific vulnerable groups like divorced and widowed women. Further study of the age distribution of pre-cancerous cervical lesions is merited.

## Supporting information

**S1 Dataset.**
(XLSX)

## Acknowledgments

Our great appreciation goes to healthcare workers of the health facilities involved in the data collection and all study participants of the study. Moreover we would like to extend our acknowledgment for administrative and community health care workers for their support during data collection and facilitation of the project.

## Author Contributions

**Conceptualization:** Gerezgiher B. Abera.

**Data curation:** Haftamu Hailekiros, Tsehaye Asmelash.

**Formal analysis:** Gerezgiher B. Abera, Henock G. Yebyo.

**Funding acquisition:** Gerezgiher B. Abera, Haftamu Hailekiros, Goitom Gigar, Tsehaye Asmelash.

**Investigation:** Gerezgiher B. Abera.

**Methodology:** Gerezgiher B. Abera, Henock G. Yebyo, Haftamu Hailekiros, Yibrah Berhe, Goitom Gigar, Tsehaye Asmelash, Gelila Goba.

**Project administration:** Gerezgiher B. Abera, Yibrah Berhe, Goitom Gigar, Tsehaye Asmelash.

**Resources:** Selam Niguse.

**Software:** Gerezgiher B. Abera.

**Supervision:** Gerezgiher B. Abera, Selam Niguse, Yibrah Berhe, Gelila Goba.

**Validation:** Gerezgiher B. Abera, Gelila Goba.

**Visualization:** Gerezgiher B. Abera.

**Writing – original draft:** Gerezgiher B. Abera.

**Writing – review & editing:** Henock G. Yebyo, Goitom Gigar, Gelila Goba.

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
