## [Decision Letter · Decision Letter 0]

4 Dec 2020

PONE-D-20-24267

Epidemiology of precancerous cervical lesion and risk factors among adult women in Tigray, Ethiopia

PLOS ONE

Dear Dr. Gerezgiher Buruh Abera:

Thank you for submitting your manuscript to PLOS ONE. After careful consideration, we feel that it has merit but does not fully meet PLOS ONE’s publication criteria as it currently stands. Therefore, we invite you to submit a revised version of the manuscript that addresses the points raised during the review process.

We look forward to receiving your revised manuscript.

Kind regards,

Joseph K.B. Matovu, Ph.D.

Academic Editor

PLOS ONE

Journal Requirements:

2. In your Methods section, please provide additional information about the participant recruitment method and the demographic details of your participants. Please ensure you have provided sufficient details to replicate the analyses such as: a) a description of any inclusion/exclusion criteria that were applied to participant recruitment, and b) a statement as to whether your sample can be considered representative of a larger population.

3. Please provide additional details regarding participant consent. In the ethics statement in the Methods , please ensure that you have specified (1) whether consent was informed and (2) what type you obtained (for instance, written or verbal, and if verbal, how it was documented and witnessed).

4. You indicated that you had ethical approval for your study. In your Methods section, please ensure you have also stated whether you obtained consent from parents or guardians of the minors included in the study or whether the research ethics committee or IRB specifically waived the need for their consent.

5. Please include additional information regarding the survey or questionnaire used in the study and ensure that you have provided sufficient details that others could replicate the analyses. For instance, if you developed a questionnaire as part of this study and it is not under a copyright more restrictive than CC-BY, please include a copy, in both the original language and English, as Supporting Information.

6. Please include a discussion of the limitations of your study in the Discussion section of your manuscript.

Additional Editor Comments (if provided):

Editor's comments to the authors

Title: Epidemiology of precancerous cervical lesion and risk factors among adult women in Tigray, Ethiopia

1. In general, I agree that there are grammatical errors throughout the paper. I think the authors may find it helpful to consult an English Language Specialist or a native English Language speaker to edit the entire paper.

2. The covering letter is not well structured. The authors should note that this is a ‘letter’ to the Editor; as such, it should be structured as a letter, addressed to the Editor, PLoS ONE, and signed off by the corresponding author. It should be clearly dated and state why the paper should be considered by the journal.

3. The authors should write in the language of research.

* “pre-cancer lesion” – should be written as ‘precancerous lesion’

* “Divorced or widowed women had risked 2.5 and 4.7 times more likely to be positive and suspicious” – the language used: ‘…women had risked 2.5 and 4.7 times…’ is not proper research language.

* “… were associated with suspicious for cancer” – associated with suspicious for cancer is not proper English

* “Data was 95 collected from March 2016 to June 2017”. The word ‘data’ is plural; the authors should use ‘were’ after the word ‘data’

4. The authors write: ‘Data were collected using an interviewer administered questionnaire…’ What data were actually collected? This is neither provided in the abstract nor in the main text of the paper.

5. The authors write, ‘We added 10% to the calculated sample size to account for non-responses’. I don’t think it is just ‘adding’ for the sake of adding. Can the authors explain why they adjusted the sample size by 10% and not any other percentage? Can they include a citation to back this up? How was the decision to adjust the sample size by 10% reached?

6. The authors should provide a little more detail on how the multistage sampling procedures were conducted. The information provided is not sufficient to explain what exactly happened at each stage.

7. In Table 4, the authors should include n/N to guide interpretation of the findings from the bivariate and multivariable models. Besides, in principle, the primary outcome is not included in the table. So, I am surprised that Table 4 includes a column for ‘VIA result’. In my view, this inclusion points to a serious error in the way the regression models were constructed.

8. The ARRR (95%CI) column within Table 4 is not well structured. The authors should ensure that all results are visible to the reader to aid interpretation.

9. The authors should ensure that all references are written in line with the journal’s referencing style and there should be consistency in the way all the references are presented.

Reviewers' comments:

Reviewer's Responses to Questions

**Comments to the Author**

1. Is the manuscript technically sound, and do the data support the conclusions?

Reviewer #1: No

Reviewer #2: Partly

2. Has the statistical analysis been performed appropriately and rigorously? 

Reviewer #1: No

Reviewer #2: I Don't Know

3. Have the authors made all data underlying the findings in their manuscript fully available?

Reviewer #1: No

Reviewer #2: No

4. Is the manuscript presented in an intelligible fashion and written in standard English?

Reviewer #1: No

Reviewer #2: No

5. Review Comments to the Author

Reviewer #1: Abstract

1. Rewrite this sentence -Divorced or widowed women had risked 2.5 and 4.7 times more 35 likely to be positive for pre-cancer lesion and suspicious, respectively, compared to single women (rrr=2.5, 95% CI [1.13, 5.52]); (rrr=4.69, 95% CI [1.00, 21.84]).

2. How is the formal education suspicious result is (rrr=0.32 below zero , and 95% CI [1.00, 21.84]) is above one? Please justify it?

3. The associated risk factor is pre-cancer lesion or/and suspicious, please clear it.

generally -the sentence is incomplete and incorrect and grammatically wrong

the analysis is wrong .

Reviewer #2: Summary: In their manuscript, Abera and colleagues present the findings of a study assessing the prevalence of precancerous cervical lesions and associated risk factors in Tigray, Ethiopia. Based on multistage sampling they recruited 900 women from the Tigray region, who then underwent visual inspection with acetic acid (VIA) and completed a questionnaire. The study question is interesting, as limited data on prevalence on cervical pre-cancer in Ethiopia are available. However, some aspects of the methodology and the results need to be clarified. Furthermore, the article would benefit from some language editing to improve comprehensibility.

Please find below specific comments for each section.

Data availability:

1. The authors state that the data are fully available and included in the manuscript. Is there an appendix with the individual patient data, or where can they be found?

Abstract:

1. In the background section the authors state that cervical cancer is the leading cause of mortality in women in LMIC. Are they referring to cancer-related mortality, or overall mortality?

2. The study aim was to assess the magnitude of precancerous lesions. May be better to use the term “prevalence” instead of “magnitude”, as magnitude could also refer to the size of the lesions.

3. In the methods part the authors state that they estimated predictors. Predictors for what?

4. The abbreviation rrr should be introduced.

5. In the conclusion, please state the implications of your findings.

Introduction:

1. Page 3, line 49: It’s important to highlight that CC is preventable only if treated early (not only detected early).

2. Page 3, line 52: Reference 5 does not fit that statement well. Rather cite Globocan?

3. Page 3, line 54: Which host factors are the authors referring to?

4. Reference 14 seems incomplete. Can this report be found online?

Methods:

1. Page 5, line 102: Why did the authors assume a cervical precancer prevalence of 50% for the sample size calculation when a previous institution-based study had found a prevalence of 6.7% in the Tigray region?

2. What were the eligibility criteria for women to participate?

3. How did the systematic random sampling work? The process should be described in more detail. Under “data collection procedure” it sounds like a convenience sample from the selected sub-districts was used.

4. Please state which variables were included in the multivariable model finally.

Results:

1. How did the authors determine the categorization of the variables, e.g. of parity (0-4. 5-12 children) and monthly income? Some of the categories contain few participants.

2. How was history of STI determined? Symptoms? Treated STI?

3. Why was history of cervical cancer screening not included as a potential predictor? It would have been interesting to see how many women had been screened previously, and whether that was associated with having cervical precancer.

4. Table 4: Please report p-values for whole variables not individual categories of variables.

Discussion:

1. Page 12, line 232: HIV status mentioned here, but not in the results section?

2. The authors should discuss the limitations of their study.

3. The discussion is mainly a comparison with results from other studies. It would be helpful if the authors could also expand on hypotheses and explanations of why certain factors may predict cervical precancer. For example, why would divorced and widowed women be at increased risk of precancer after adjustment for age? Any hypotheses?

6. PLOS authors have the option to publish the peer review history of their article (what does this mean?). If published, this will include your full peer review and any attached files.

Reviewer #1: No

Reviewer #2: No

---

## [Author Response · Author response to Decision Letter 0]

1 Jun 2021

To: PLOSOne Editor-in-Chief 

Subject: -Point by point response for the comments and recommendations of our reviewers 

Title: “Epidemiology of precancerous cervical lesion and risk factors among adult women in Tigray, Ethiopia."

Reference: PONE-D-20-24267

Dear Editor-in-Chief

We are very grateful for the consideration of the manuscript. In accordance with the reviewers’ valuable comments and recommendations, we have revised the manuscript and we hereby submit the revised work for your consideration.

Comments from the Editor:

• Response: This manuscript is prepared based on the information for author format from website of PLOSOne as much as possible

2. In your Methods section, please provide additional information about the participant recruitment method and the demographic details of your participants. Please ensure you have provided sufficient details to replicate the analyses such as: a) a description of any inclusion/exclusion criteria that were applied to participant recruitment, and b) a statement as to whether your sample can be considered representative of a larger population.

• Response: based on your valuable comment, we made clarification about the recruitment procedure, mentioned under the method section of the main document. 

• The eligibility criteria specify who to include in the study. As mentioned under the multistage sampling technique, areas and participants were allocated proportionally and were selected using probability sampling method. We assume that this selection approach suffice the representativeness of the study participants.

3. Please provide additional details regarding participant consent. In the ethics statement in the Methods , please ensure that you have specified (1) whether consent was informed and (2) what type you obtained (for instance, written or verbal, and if verbal, how it was documented and witnessed).

• Response: We accept the comment and written informed consent was obtained as mentioned under ethical consideration subsection of the method section.

4. You indicated that you had ethical approval for your study. In your Methods section, please ensure you have also stated whether you obtained consent from parents or guardians of the minors included in the study or whether the research ethics committee or IRB specifically waived the need for their consent.

• Responses: Since most participants were village dwellers, married and live independently, those above 15 years married and live independently were considered to sign the consent by their selves. This was mentioned in the protocol before obtaining the ethical clearance.

5. Please include additional information regarding the survey or questionnaire used in the study and ensure that you have provided sufficient details that others could replicate the analyses. For instance, if you developed a questionnaire as part of this study and it is not under a copyright more restrictive than CC-BY, please include a copy, in both the original language and English, as Supporting Information.

• Response: We agree with your concern and we can assure that we prepared the questionnaire from different literatures. We can submit our questionnaire up on request. 

6. Please include a discussion of the limitations of your study in the Discussion section of your manuscript.

• Response: We accept the comment and correction was made in the main text of the discussion section.

Additional Editor Comments (if provided):

Editor's comments to the authors

Title: Epidemiology of precancerous cervical lesion and risk factors among adult women in Tigray, Ethiopia

1. In general, I agree that there are grammatical errors throughout the paper. I think the authors may find it helpful to consult an English Language Specialist or a native English Language speaker to edit the entire paper.

• Response: We accept the comment and we have received a copy edit of English language from English language professional.

2. The covering letter is not well structured. The authors should note that this is a ‘letter’ to the Editor; as such, it should be structured as a letter, addressed to the Editor, PLoS ONE, and signed off by the corresponding author. It should be clearly dated and state why the paper should be considered by the journal.

• Response: We accept the comment and we have made revision of the cover.

3. The authors should write in the language of research.

* “pre-cancer lesion” – should be written as ‘precancerous lesion’

* “Divorced or widowed women had risked 2.5 and 4.7 times more likely to be positive and suspicious” – the language used: ‘…women had risked 2.5 and 4.7 times…’ is not proper research language.

* “… were associated with suspicious for cancer” – associated with suspicious for cancer is not proper English

* “Data was 95 collected from March 2016 to June 2017”. The word ‘data’ is plural; the authors should use ‘were’ after the word ‘data’

• Response: We accept the comments above and we have made corrections in the abstract and pages 5 and 12 in the main txt.

4. The authors write: ‘Data were collected using an interviewer administered questionnaire…’ What data were actually collected? This is neither provided in the abstract nor in the main text of the paper.

• Response: The content of the questionnaire comprises information on socio-demographic and risk-factors, including health related behaviors, lifestyle, knowledge, attitude, and clinical attributes With due resection, we can assure that it has been mentioned under data collection section, in the previous version

5. The authors write, ‘We added 10% to the calculated sample size to account for non-responses’. I don’t think it is just ‘adding’ for the sake of adding. Can the authors explain why they adjusted the sample size by 10% and not any other percentage? Can they include a citation to back this up? How was the decision to adjust the sample size by 10% reached?

• Response: We accept that it needs justification. Sample size determination gives a minimum possible sample and maximizing sample size provides good representativeness, while more cot. After we determine the sample size, we assume that there will be non-respondents because of the cervical screening procedure using VIA. But we did not know how many of the recruited participants may refuse the procedure. Therefore 10% as compared to 5% would better compensate the number of non-respondents. 

6. The authors should provide a little more detail on how the multistage sampling procedures were conducted. The information provided is not sufficient to explain what exactly happened at each stage.

• Response: We accept the comments and we have made corrections under the sampling technique section of the main text.

7. In Table 4, the authors should include n/N to guide interpretation of the findings from the bivariate and multivariable models. Besides, in principle, the primary outcome is not included in the table. So, I am surprised that Table 4 includes a column for ‘VIA result’. In my view, this inclusion points to a serious error in the way the regression models were constructed.

• Response: The analysis model used is multinomial logistic regression with “Negative” result as base reference. Based on your valuable recommendation, we have made correction in the format of table 4. 

8. The ARRR (95%CI) column within Table 4 is not well structured. The authors should ensure that all results are visible to the reader to aid interpretation.

• Response: Based on your valuable recommendation, we have made correction in the format of table 4. 

9. The authors should ensure that all references are written in line with the journal’s referencing style and there should be consistency in the way all the references are presented.

• Response: We accept the comment and all published journals are written in Vancouver style as per the PLOSOne information for authors. Reference number 23 is used in the method section page 8.

Reviewers' comments:

Reviewer's Responses to Questions

Comments to the Author

1. Is the manuscript technically sound, and do the data support the conclusions?

Reviewer #1: No

Reviewer #2: Partly

2. Has the statistical analysis been performed appropriately and rigorously? 

Reviewer #1: No

Reviewer #2: I Don't Know

3. Have the authors made all data underlying the findings in their manuscript fully available?

Reviewer #1: No

Reviewer #2: No

4. Is the manuscript presented in an intelligible fashion and written in standard English?

Reviewer #1: No

Reviewer #2: No

5. Review Comments to the Author

Reviewer #1: Abstract

1. Rewrite this sentence -Divorced or widowed women had risked 2.5 and 4.7 times more 35 likely to be positive for pre-cancer lesion and suspicious, respectively, compared to single women (rrr=2.5, 95% CI [1.13, 5.52]); (rrr=4.69, 95% CI [1.00, 21.84]).

• Response: We accept the comment and correction has been made under abstract and result sections.

2. How is the formal education suspicious result is (rrr=0.32 below zero , and 95% CI [1.00, 21.84]) is above one? Please justify it?

• Response: Dear/Sir/Madam, if we did not mistaken, rrr = 0.32 is for the Primary Education, while and 95% CI [1.00, 21.84] is for divorced or widowed variable. The C.I. for primary education variable with rrr = 0.32 is 0.32(0.11, 0.91), which does not include 1.

3. The associated risk factor is pre-cancer lesion or/and suspicious, please clear it.

generally -the sentence is incomplete and incorrect and grammatically wrong

the analysis is wrong .

• Response: We accept the comment and we have a concern on that VIA could not diagnose a cancer but it is good to understand that precancerous cervical lesion screening will come up with the results of negative, positive, and suspicious for cancer. In this sense, auspicious means not a cancer but pre-cancer till confirmed with diagnosis. 

Reviewer #2: Summary: In their manuscript, Abera and colleagues present the findings of a study assessing the prevalence of precancerous cervical lesions and associated risk factors in Tigray, Ethiopia. Based on multistage sampling they recruited 900 women from the Tigray region, who then underwent visual inspection with acetic acid (VIA) and completed a questionnaire. The study question is interesting, as limited data on prevalence on cervical pre-cancer in Ethiopia are available. However, some aspects of the methodology and the results need to be clarified. Furthermore, the article would benefit from some language editing to improve comprehensibility.

Please find below specific comments for each section.

Data availability:

1. The authors state that the data are fully available and included in the manuscript. Is there an appendix with the individual patient data, or where can they be found?

• Response: only data in table 1, 2, and 3 were collected in this paper. Except the identifiers (like ID), there is no additional information collected than tables 1, 2, and 3. If the format of the empty questionnaire is needed, we can upload when requested.

1. In the background section the authors state that cervical cancer is the leading cause of mortality in women in LMIC. Are they referring to cancer-related mortality, or overall mortality?

• Response: We accept the comment and correction is made in the main text

2. The study aim was to assess the magnitude of precancerous lesions. May be better to use the term “prevalence” instead of “magnitude”, as magnitude could also refer to the size of the lesions.

• Response: We accept the comment and correction is made in the main text

3. In the methods part the authors state that they estimated predictors. Predictors for what?

• Response: We accept the comment and it is to mention the predictors for cervical precancerous lesion.

4. The abbreviation rrr should be introduced.

• Response: The abbreviation rrr has been introduced in the “list of abbreviations” section.

5. In the conclusion, please state the implications of your findings.

• Response: We accept the comment and correction is made accordingly

Introduction:

1. Page 3, line 49: It’s important to highlight that CC is preventable only if treated early (not only detected early).

• Response: We accept the comment and correction is made under abstract and introduction sections

2. Page 3, line 52: Reference 5 does not fit that statement well. Rather cite Globocan?

• Response: We accept the comment and after checking the references, correction is made for the references

3. Page 3, line 54: Which host factors are the authors referring to?

• Response: We accept the comment and we refer the participants’ factor as the host factors.

4. Reference 14 seems incomplete. Can this report be found online?

• Response: We accept the comment and correction is made. It is an annual report filed as government document. 

Methods:

1. Page 5, line 102: Why did the authors assume a cervical precancer prevalence of 50% for the sample size calculation when a previous institution-based study had found a prevalence of 6.7% in the Tigray region?

• Response: In Ethiopia, VIA screening was not available for non HIV women until 2015. It was only three hospitals providing VIA screening for HIV patents in the stufdy area, Tigray. Therefore the above mentioned prevalence is for HIV patients. Moreover, the above prevalence would provide us less number of participants and it is advisable to increases sample size if affordable. Therefore, we assume increasing participants would improve the representativeness of the eligible participants

.

2. What were the eligibility criteria for women to participate?

• Response: It was not elaborated earlier, that all adult women, who were sexually active, voluntarily consented to participate, and those did not undergo hysterectomy were the inclusion criteria, as now mentioned in the main text.

3. How did the systematic random sampling work? The process should be described in more detail. Under “data collection procedure” it sounds like a convenience sample from the selected sub-districts was used.

• Response: Dear/Sir/Madam: about 3500 participants were estimated to attend the screening campaign (in fact 3866 were attending the screening campaign). Based on this source population, a key value of 4 was calculated (3500/900). Based on their arrival every forth was invited to participate in the study. 

4. Please state which variables were included in the multivariable model finally.

• Response: Dear/Sir/Madam: the model variables are mentioned under the “factors associated---,” sub section, in paragraph two.

Results:

1. How did the authors determine the categorization of the variables, e.g. of parity (0-4. 5-12 children) and monthly income? Some of the categories contain few participants.

• Response: Related to the parity, in Ethiopia fertility rate is 4.4; hence we categorized those below fertility rate and above fertility rate. But for the income category, though we could merge it, in the suspicious category it is still below 5% because of the suspicious cases are few. Therefore, if it is advisable, we think it will be better to remove up on recommendation. 

2. How was history of STI determined? Symptoms? Treated STI?

• Response: We determine the STI history based on items related to symptom of the infections taken from the national syndrome approach of the STI guideline. 

3. Why was history of cervical cancer screening not included as a potential predictor? It would have been interesting to see how many women had been screened previously and whether that was associated with having cervical precancer.

• Response: we respect your valuable comment. First of all, there was no routine cervical screening mechanism in the country; except for HIV cases started 2011. In the absence of service, asking utilization is not logical.

4. Table 4: Please report p-values for whole variables not individual categories of variables.

• Response: We accept the concern, but this may not specify which category or level of a variable is associated with the outcome variable.

Discussion:

1. Page 12, line 232: HIV status mentioned here, but not in the results section?

• Response: We accept the comment and correction is made accordingly

2. The authors should discuss the limitations of their study.

• Response: We accept the comment and correction is made accordingly

3. The discussion is mainly a comparison with results from other studies. It would be helpful if the authors could also expand on hypotheses and explanations of why certain factors may predict cervical precancer. For example, why would divorced and widowed women be at increased risk of precancer after adjustment for age? Any hypotheses?

• Response: We accept the comment and correction is made accordingly

6. PLOS authors have the option to publish the peer review history of their article (what does this mean?). If published, this will include your full peer review and any attached files.

Do you want your identity to be public for this peer review? For information about this choice, including consent withdrawal, please see our Privacy Policy.

Reviewer #1: No

Reviewer #2: No

---

## [Decision Letter · Decision Letter 1]

5 Jul 2021

PONE-D-20-24267R1

Epidemiology of Cervical Precancerous Lesion and Risk Factors Among Adult Women In Tigray, Ethiopia

PLOS ONE

Dear Dr. Abera,

Thank you for submitting your manuscript to PLOS ONE. After careful consideration, we feel that it has merit but does not fully meet PLOS ONE’s publication criteria as it currently stands. Therefore, we invite you to submit a revised version of the manuscript that addresses the points raised during the review process.

We look forward to receiving your revised manuscript.

Kind regards,

Joseph K.B. Matovu, Ph.D.

Academic Editor

PLOS ONE

Additional Editor Comments (if provided):

The authors have tried to address some of my comments but the paper still requires a lot of work before a decision can be made. I would like to request the authors to take time to think about the comments raised and try to offer appropriate responses to ALL the comments raised. We can't continue to return the same comments forever. So, the authors should look at the comments returned by reviewer No. 2 as well as myself and endeavor to fully address all the areas that were inadequately addressed. For instance, I raised issues on the grammatical errors which the authors indicated were addressed but to the contrary, the same errors as before still exist in the paper. I had a comment on the formatting of the references and requested the authors to revise them according to the journal's formatting requirements. However, the references were returned with the same errors as before. Besides, the explanation provided for adjusting the sample size by 10% is not sufficient. What is needed is to include a citation that backs up the use of this percentage. This was not done. Here are other additional comments for the authors to consider.

1. The paper STILL has many grammatical errors that would benefit from being reviewed by an English Specialist. I raised similar comments earlier but these aspects are not yet fully addressed. For instance, within the abstract, instead of writing, ‘*Divorced or widowed women had risked 2.5 and 4.7 times more likely to be positive and suspicious, respectively, compared to single women…*’, the authors should write: ‘Divorced or widowed women were 2.5 and 4.7 times more likely to be positive and suspicious… than single women’ The expressed: “had risked” is wrongly used. In addition, the use of ‘more likely’ begs the use of ‘than’ rather than ‘compared to’. Similar statements exist in the abstract and in the main manuscript that should be addressed.

Please also fix the following errors/issues in the abstract:

*Expand ‘rrr’ at first mention, and then abbreviate thereafter

*The word ‘In’ in the title should be changed to ‘in’, while ‘Among’ should be written as ‘among’.

*95%CI should be written in expanded form at first mention, and then abbreviate thereafter

*’*…68% lower for women with primary education compared to those with no formal education*’ – use ‘than’ instead of ‘compared to’.

*’*…History of sexually transmitted infection was associated with positive pre-cancer lesion*’ – change ‘pre-cancer lesion’ to ‘pre-cancerous cervical lesion’.

*’*Seventy-nine (8.95%) women were positive for pre-cancer lesion*’ – “pre-cancer lesion” should be written as ‘pre-cancerous cervical lesions’

*’*…were associated with suspicious for cancer*’ – rewrite as: ‘… were associated with having a suspicious cervical cancer result’

*In the title, the authors refer to ‘risk factors’ but in the conclusion, they refer to ‘predictors’. Do these two terms mean the same?

2. In the abstract, the expression: ‘*A community-based cross sectional study was used*…’ should be written as: ‘This was a community-based, cross-sectional study conducted among 900 women in Tigray region, Ethiopia, from --- to ----‘. In other words, the opening statement on the ‘Methods’ sub-section should tell the reader what the study design was, where the study was conducted, and when the study was done. Please note the use of the ‘comma’ after the word ‘community-based’ and the ‘hyphen’ inserted in the words ‘cross sectional’ to turn them into one word.

3. General formatting requirements. The authors should ensure that they follow the general formatting requirements of the journal. For instance, the formatting for the sections indicates how the headings and sub-headings should be formatted, etc. For instance, under the ‘Methods’ section (the authors write it as ‘Method’), this should be formatted as:

**Materials and Methods **(Level 1: bold type; 18pt font)

xxx

**Study setting and design** (Level 2: for sub-sections of major sections; Bold type, 16pt font)

4. Other corrections

*Use of abbreviations: Please ensure that ALL abbreviations are presented in expanded form at first mention in the abstract or body of the manuscript. Even if these abbreviations have been listed in the list of abbreviations at the end of the paper, this does not take away the need to expand all abbreviations at first mention.

*Line 153, page 7: ‘*Data was analysed*…’ Data is a plural word; so, revise as: ‘Data were analysed…’

*Lines 163-164, page 8: the authors write: ‘*…ethicalclearance was obtained from mekelle university, collage of health sience research and community service commiyyee*’. The word ‘ethicalclearance’ should be written as two words (ethical clearance). Names of places should always begin with a capital letter, e.g. Mekelle University… The word ‘sience’ should be edited to ‘Sciences’. This also applies to ‘collage’ which should be written as ‘College’. The word ‘commiyyee’ does not exist in English. I think the authors meant to write: ‘committee’. In general, the entire paper should be edited to address any other areas that are not included in this report. The entire sentence should be revised as follows:

‘**… ethical clearance was obtained from Mekelle University College of Health Sciences Research and Community Service Committee …**’

*Please note that should include the study protocol clearance numbers (e.g. Protocol#: 001/2019’) as per your country’s national research management guidelines.

*Ensure consistency in writing the word ‘pre-cancerous’. In some sections of the paper, the word is written as ‘precancerous’ while in others, it is written as ‘pre-cancerous’. Please choose one form and maintain it throughout the paper.

*Check ALL the references used against the journal’s referencing guidelines. I can see that some journal names are italicized while others are not; I can also see that the referenced paper’s title ends with a comma instead of a full stop. Other references have the journal name written in full. Please check the journal’s referencing style (https://journals.plos.org/plosone/s/submission-guidelines) and fix all the errors in the reference section. When you click on the link, scroll down and look for ‘references’. There is a table that shows you how different references should be written.

*Reference 15 reads: ‘*Dye D et al., 2009) Dye D., Solomon B., Claire H., Yared T., Vanessa H., Teshome D., 384 Marion B., Anne R., (2009) Complex care systems in developing countries. Breast 385 Cancer Patient Navigation in Ethiopia 116:577–85.)*’ This is an example of bad referencing style. In general, please ensure that the references are well formatted based on the journal style. Please note that:

- Where there are more than six authors, please cite three of them followed by the word ‘et al.’

- Where there are six authors or less, list all the authors.

- A reference cannot have two years of publication in the same reference, as shown in the example above

- The year of publication should come at the end, after the journal’s name

- PLoS ONE does not use the author version (Anne R., or Marion B., ...). Click on the link above for guidance on how to write authors' names in the references section.

- PLoS ONE uses abbreviated journal names. Please ensure that the journal names are abbreviated but not italicized.

5. Supplementary information. The journal requires that: “**Authors can submit essential supporting files and multimedia files along with their manuscripts. All supporting information will be subject to peer review. All file types can be submitted, but files must be smaller than 20 MB in size**.” I did not see any supplementary information provided. Please click on the link above and check for ‘supplementary information’. Here are examples of supplementary information that should be included:

*Data collection tools

*Dataset used during the analysis

* Please check the journal’s guidelines on how supplementary information should be formatted.

6. Additional information requested at submission: All manuscripts should carry the following sub-section (please click on the link above for guidance on how to include this sub-section):

- Authors’ contributions

Reviewers' comments:

Reviewer's Responses to Questions

**Comments to the Author**

1. If the authors have adequately addressed your comments raised in a previous round of review and you feel that this manuscript is now acceptable for publication, you may indicate that here to bypass the “Comments to the Author” section, enter your conflict of interest statement in the “Confidential to Editor” section, and submit your "Accept" recommendation.

Reviewer #1: All comments have been addressed

Reviewer #2: (No Response)

2. Is the manuscript technically sound, and do the data support the conclusions?

Reviewer #1: Yes

Reviewer #2: No

3. Has the statistical analysis been performed appropriately and rigorously? 

Reviewer #1: Yes

Reviewer #2: No

4. Have the authors made all data underlying the findings in their manuscript fully available?

Reviewer #1: Yes

Reviewer #2: No

5. Is the manuscript presented in an intelligible fashion and written in standard English?

Reviewer #1: Yes

Reviewer #2: No

6. Review Comments to the Author

Reviewer #1: (No Response)

Reviewer #2: Summary: The comments have been incompletely addressed, and there are still numerous spelling/grammatical errors.

Please find below specific comments for each section.

Data availability:

1. Previous comment: The authors state that the data are fully available and included in the manuscript. Is there an appendix with the individual patient data, or where can they be found?

Author reply: only data in table 1, 2, and 3 were collected in this paper. Except the

identifiers (like ID), there is no additional information collected than tables 1, 2, and 3.

If the format of the empty questionnaire is needed, we can upload when requested.

Comment: Data in table 1,2, and 3 are aggregate data, not individual patient data. It is not possible to replicate the analyses with these data. Can the dataset with the individual patient data be accessed somewhere? If not, the data availability statement needs to be addressed accordingly.

Abstract:

1. Previous comment: The abbreviation rrr should be introduced.

Author reply: The abbreviation rrr has been introduced in the “list of abbreviations”

section.

Comment: All abbreviations should be introduced in the manuscript text, not only in the list of abbreviations.

2. Previous comment: In the conclusion, please state the implications of your findings.

Author reply: We accept the comment and correction is made accordingly.

Comment: What changes were made? I don’t see any in the conclusion.

Introduction:

1. Previous comment: Page 3, line 54: Which host factors are the authors referring to?

Author reply: We accept the comment and we refer the participants’ factor as the host

factors.

Comment: It is not enough to refer to participants’ factors instead of host factors. Which participant factors may be relevant?

Methods:

1. Previous comment: Page 5, line 102: Why did the authors assume a cervical precancer prevalence of 50% for the sample size calculation when a previous institution-based study had found a prevalence of 6.7% in the Tigray region?

Author reply: In Ethiopia, VIA screening was not available for non HIV women until

2015. It was only three hospitals providing VIA screening for HIV patents in the stufdy

area, Tigray. Therefore the above mentioned prevalence is for HIV patients. Moreover,

the above prevalence would provide us less number of participants and it is advisable

to increases sample size if affordable. Therefore, we assume increasing participants

would improve the representativeness of the eligible participants.

Comment: I do not understand this reply. Women living with HIV are at higher risk of developing precancerous cervical lesions. If the estimate of 6.7% stems from a study among women living with HIV, then a lower prevalence should be assumed for the reported study including HIV-negative women.

2. Previous comment: Please state which variables were included in the multivariable model finally.

Author reply: Dear/Sir/Madam: the model variables are mentioned under the “factors

associated---,” sub section, in paragraph two.

Response: From that paragraph it is not clear which variables were included in the final multivariable model. Furthermore, the authors should explain how they chose variables for inclusion in the multivariable model. This information should be given in the methods section.

3. Eligibility criteria: “all adult women” – please state an age range. Were women eligible for screening from the age of 15 already? What do the national cervical cancer screening guidelines recommend?

Results:

1. Previous comment: How was history of STI determined? Symptoms? Treated STI?

Author response: We determine the STI history based on items related to symptom of the

infections taken from the national syndrome approach of the STI guideline.

Comment: This information should be added to the manuscript.

2. Previous comment: Why was history of cervical cancer screening not included as a potential predictor? It would have been interesting to see how many women had been screened previously, and whether that was associated with having cervical precancer.

Author response: we respect your valuable comment. First of all, there was no routine

cervical screening mechanism in the country; except for HIV cases started 2011. In

the absence of service, asking utilization is not logical.

Comment: This background information should be added to the manuscript as readers may not be aware of this.

3. Table 4: For some of the variables (e.g., age) there is no reference category for aRRR. The analysis looks wrong.

4. Table 4: explanations for *, **, and *** are missing.

Discussion:

1. Previous comment: Page 12, line 232: HIV status mentioned here, but not in the results section?

Author response: We accept the comment and correction is made accordingly.

Comment: What is the correction? That HIV status is removed entirely? Does that mean that no information was available on HIV status?

2. Previous comment: The authors should discuss the limitations of their study.

Author response: We accept the comment and correction is made accordingly.

Comment: Discussion of study limitations is insufficient.

3. Previous comment: The discussion is mainly a comparison with results from other studies. It would be helpful if the authors could also expand on hypotheses and explanations of why certain factors may predict cervical precancer. For example, why would divorced and widowed women be at increased risk of precancer after adjustment for age? Any hypotheses?

Author response: We accept the comment and correction is made accordingly.

Comment: This comment has been inadequately addressed, and the discussion section is difficult to understand sometimes. E.g., “In Ethiopia, long time contraceptive users are few, which is evidenced by having high fertility rate. This may not predispose to cervical precancerious lesion.” – what does that mean? What does not predispose to precancerous cervical lesions? The discussion section needs to be restructured and improved.

7. PLOS authors have the option to publish the peer review history of their article (what does this mean?). If published, this will include your full peer review and any attached files.

Reviewer #1: No

Reviewer #2: No

---

## [Author Response · Author response to Decision Letter 1]

1 Jul 2022

Date: February 12 2022

To: PLOS ONE" plosone@plos.org

From: Gerezgiher Buruh Abera" gbamsc2002@gmail.com (PI)

Subject: Point by Point response to editor and reviewer(s).

Title Epidemiology of Pre-cancerous Cervical Lesion and Risk Factors among Adult Women in Tigray, Ethiopia

Reference PONE-D-20-24267R1

Dear/Sir/ professors: editor and reviewer: 

We apologize for the late response! It is because of the internet lockdown in Tigray, North Ethiopia, due to crises. We are responding through the internet from NGOs that allow to use for publications, international projects and PhD students only. We are very grateful for the consideration of the manuscript. In accordance with the editors and reviewers’ valuable comments and recommendations, we have revised the manuscript and we are hereby submitting the revised work for your consideration. Pages referred in this letter are based on the manuscript with track change.

Additional Editor Comments (if provided):

The authors have tried to address some of my comments but the paper still requires a lot of work before a decision can be made. I would like to request the authors to take time to think about the comments raised and try to offer appropriate responses to ALL the comments raised. We can't continue to return the same comments forever. So, the authors should look at the comments returned by reviewer No. 2 as well as myself and endeavor to fully address all the areas that were inadequately addressed. For instance, I raised issues on the grammatical errors which the authors indicated were addressed but to the contrary, the same errors as before still exist in the paper. I had a comment on the formatting of the references and requested the authors to revise them according to the journal's formatting requirements. However, the references were returned with the same errors as before. Besides, the explanation provided for adjusting the sample size by 10% is not sufficient. What is needed is to include a citation that backs up the use of this percentage. This was not done. Here are other additional comments for the authors to consider.

• Response: Sorry for the inconvenience! We were not to refuse comments. To be genuine, I am not competent in English, since it is my second language. In the previous time, we provide it to colleagues in Illinois University, a native English speaker (through the last listed co-author in this manuscript, She is from Chicago), and at that time we received the copy of the language edited version. Now we tried to go through the manuscript to correct grammatical errors as much as possible based on the direction given.

• We accept the comment on the format of the reference and we tried to correct especially reference 15 based on the old version of the paper, but we could not able to Google to recheck the citation style.

• A different contingency are commonly used from 5% to up to 30% (as in trials), based on the study design. In our cross sectional study, VIA screening was used, which is among the factors that could increase nonresponse rate. Now, we had added sample references to our manuscript. Based on previous studies, response rates ranges from 77.1% (24) to 98.2%, (25). This implies a non-response rate of 1.8% to 29.9% Therefore, based on the average of the references; we assume that 10% would enough to maximize the sample size to compensate the non-response rate expected. 

1. The paper STILL has many grammatical errors that would benefit from being reviewed by an English Specialist. I raised similar comments earlier but these aspects are not yet fully addressed. For instance, within the abstract, instead of writing, ‘Divorced or widowed women had risked 2.5 and 4.7 times more likely to be positive and suspicious, respectively, compared to single women…’, the authors should write: ‘Divorced or widowed women were 2.5 and 4.7 times more likely to be positive and suspicious… than single women’ The expressed: “had risked” is wrongly used. In addition, the use of ‘more likely’ begs the use of ‘than’ rather than ‘compared to’. Similar statements exist in the abstract and in the main manuscript that should be addressed.

• Response - We accept the comments and it is corrected accordingly, as it can be seen in abstract, page 2 and result sections.

 Please also fix the following errors/issues in the abstract:

*Expand ‘rrr’ at first mention, and then abbreviate thereafter

*The word ‘In’ in the title should be changed to ‘in’, while ‘Among’ should be written as ‘among’.

*95%CI should be written in expanded form at first mention, and then abbreviate thereafter

*’…68% lower for women with primary education compared to those with no formal education’ – use ‘than’ instead of ‘compared to’.

*’…History of sexually transmitted infection was associated with positive pre-cancer lesion’ – change ‘pre-cancer lesion’ to ‘pre-cancerous cervical lesion’.

*’Seventy-nine (8.95%) women were positive for pre-cancer lesion’ – “pre-cancer lesion” should be written as ‘pre-cancerous cervical lesions’

*’…were associated with suspicious for cancer’ – rewrite as: ‘… were associated with having a suspicious cervical cancer result’

*In the title, the authors refer to ‘risk factors’ but in the conclusion, they refer to ‘predictors’. Do these two terms mean the same?

• Response: We accept the comment and rrr is written in expanded form (relative risk ratio) under the title of “data analysis” at last paragraph. In abstract, we prefer modification, to delete the values of regression including the rrr and represents using summarized statements. This minimizes the presentation of abstract and did not compromise the meaning, since it is already described under the result in detail.

• We accept the comment and the initial letter of ‘In’ ‘and ‘Among’ in the title has been changed to small letters.

• We accept the comment and CI is written in expanded form (confidence interval) under the title of “Factors associated with cervical screening VIA result” at first paragraph of page 12.

• We accept the comment and the ‘compared to’ has been change to “than” as can be seen under the title of “Factors associated with cervical screening VIA result” at page 12, and under abstract.

• We accept the comment and the word “pre-cancerous cervical lesion’ is used in the whole manuscript to replace all miss spelled “cervical precancerous lesion.

• We accept the comment and the meaning of ‘risk factors’ (something that contributes or indicates to illness: a features or habits, or personal history that increases the probability of disease) and ‘predictors’ (something explains what is going to happen in the future, often on the basis of present). We corrected as ‘risk factors’ as it can be seen in the conclusion section of the abstract and result titles.

2. In the abstract, the expression: ‘A community-based cross sectional study was used…’ should be written as: ‘This was a community-based, cross-sectional study conducted among 900 women in Tigray region, Ethiopia, from --- to ----‘. In other words, the opening statement on the ‘Methods’ sub-section should tell the reader what the study design was, where the study was conducted, and when the study was done. Please note the use of the ‘comma’ after the word ‘community-based’ and the ‘hyphen’ inserted in the words ‘cross sectional’ to turn them into one word.

• Response: We accept the comment and corrected accordingly, as it can be seen in the abstract

3. General formatting requirements. The authors should ensure that they follow the general formatting requirements of the journal. For instance, the formatting for the sections indicates how the headings and sub-headings should be formatted, etc. For instance, under the ‘Methods’ section (the authors write it as ‘Method’), this should be formatted as:

Materials and Methods (Level 1: bold type; 18pt font)

xxx

Study setting and design (Level 2: for sub-sections of major sections; Bold type, 16pt font)

• Response: We accept the comment and the titles and subtitles has been formatted as level 1 and level 2 respectively and Method has been changed to Materials and Methods.

4. Other corrections

*Use of abbreviations: Please ensure that ALL abbreviations are presented in expanded form at first mention in the abstract or body of the manuscript. Even if these abbreviations have been listed in the list of abbreviations at the end of the paper, this does not take away the need to expand all abbreviations at first mention.

• Response: We accept the comments, and tried to correct accordingly through the whole manuscript. 

*Line 153, page 7: ‘Data was analysed…’ Data is a plural word; so, revise as: ‘Data were analysed…’

*Lines 163-164, page 8: the authors write: ‘…ethicalclearance was obtained from mekelle university, collage of health sience research and community service commiyyee’. The word ‘ethicalclearance’ should be written as two words (ethical clearance). Names of places should always begin with a capital letter, e.g. Mekelle University… The word ‘sience’ should be edited to ‘Sciences’. This also applies to ‘collage’ which should be written as ‘College’. The word ‘commiyyee’ does not exist in English. I think the authors meant to write: ‘committee’. In general, the entire paper should be edited to address any other areas that are not included in this report. The entire sentence should be revised as follows:

‘… ethical clearance was obtained from Mekelle University College of Health Sciences Research and Community Service Committee …’

• Response: We accept the comments, and it is corrected accordingly under the ‘data analysis and ethical consideration sub sections. 

*Please note that should include the study protocol clearance numbers (e.g. Protocol#: 001/2019’) as per your country’s national research management guidelines.

• Response: We accept the comments, and the registration number has been included in the manuscript, under “Ethical consideration sub section. 

*Ensure consistency in writing the word ‘pre-cancerous’. In some sections of the paper, the word is written as ‘precancerous’ while in others, it is written as ‘pre-cancerous’. Please choose one form and maintain it throughout the paper.

• Response: We accept the comments, and we tried to correct throughout the manuscript accordingly. 

*Check ALL the references used against the journal’s referencing guidelines. I can see that some journal names are italicized while others are not; I can also see that the referenced paper’s title ends with a comma instead of a full stop. Other references have the journal name written in full. Please check the journal’s referencing style (https://journals.plos.org/plosone/s/submission-guidelines) and fix all the errors in the reference section. When you click on the link, scroll down and look for ‘references’. There is a table that shows you how different references should be written.

• Response: We accept the comments. As per our understanding, some journals specify citation preference with some abbreviations (example – BMC Res Notes is for BMC research Notes e.tc.), while others not, for those journals we took as it is. Because we may miss spell it when we abbreviate ourselves. The big challenge is that I am now in Tigray, North Ethiopia where the crisis found and all way of communication are locked. I am responding this Email through NGOs, which allow us to use internet for project and PhD publication only. I am sorry to say that I could not get internet access to either research the journals or to contact my most co-authors to recheck the reference (since they are living abroad). Anyway, I had tried to address some errors as I could.

*Reference 15 reads: ‘Dye D et al., 2009) Dye D., Solomon B., Claire H., Yared T., Vanessa H., Teshome D., 384 Marion B., Anne R., (2009) Complex care systems in developing countries. Breast 385 Cancer Patient Navigation in Ethiopia 116:577–85.)’ This is an example of bad referencing style. 

• Response: We accept the comments, and we tried to correct reference 15 in the revised manuscript by referring the old version.

In general, please ensure that the references are well formatted based on the journal style. 

Please note that:

- Where there are more than six authors, please cite three of them followed by the word ‘et al.’

- Where there are six authors or less, list all the authors.

- A reference cannot have two years of publication in the same reference, as shown in the example above

- The year of publication should come at the end, after the journal’s name

- PLoS ONE does not use the author version (Anne R., or Marion B., ...). Click on the link above for guidance on how to write authors' names in the references section.

- PLoS ONE uses abbreviated journal names. Please ensure that the journal names are abbreviated but not italicized.

• Response: We accept the comments. For the name of authors in the references, it is difficult to differentiate the surname to be abbreviated, example - Amos D. Mwaka. We understand D. could be the first name. Therefore, when we use initial letter of last name first, we could mistake the first name rather than last name; for that, we prefer to use as it has been published in the journals. 

5. Supplementary information. The journal requires that: “Authors can submit essential supporting files and multimedia files along with their manuscripts. All supporting information will be subject to peer review. All file types can be submitted, but files must be smaller than 20 MB in size.” I did not see any supplementary information provided. Please click on the link above and check for ‘supplementary information’. Here are examples of supplementary information that should be included:

*Data collection tools

*Dataset used during the analysis

* Please check the journal’s guidelines on how supplementary information should be formatted.

• Response: We accept the comments, and we had uploaded the dataset in excel used during the analysis. There is no multimedia file which did not incorporated in the analysis and all of the relevant information is included in the manuscript. Data uploaded in the form of tables are used to analyze and develop the manuscript. For the confidentiality issue, we did not upload filled questionnaires. 

6. Additional information requested at submission: All manuscripts should carry the following sub-section (please click on the link above for guidance on how to include this sub-section):

- Authors’ contributions 

• Response: We accept the comments. Since it is short time allowed to use internet, we could not check the link to recheck requirements, but we used the link when we were preparing the manuscript for PLoS ONE. We understand that declaration is important and we added the declaration section as additional information including Authors’ contributions and others at the end of this manuscript above acknowledgement sub section.

Reviewers' comments:

Reviewer's Responses to Questions

Comments to the Author

1. If the authors have adequately addressed your comments raised in a previous round of review and you feel that this manuscript is now acceptable for publication, you may indicate that here to bypass the “Comments to the Author” section, enter your conflict of interest statement in the “Confidential to Editor” section, and submit your "Accept" recommendation.

Reviewer #1: All comments have been addressed

Reviewer #2: (No Response)

2. Is the manuscript technically sound, and do the data support the conclusions?

Reviewer #1: Yes

Reviewer #2: No

3. Has the statistical analysis been performed appropriately and rigorously?

Reviewer #1: Yes

Reviewer #2: No

4. Have the authors made all data underlying the findings in their manuscript fully available?

Reviewer #1: Yes

Reviewer #2: No

5. Is the manuscript presented in an intelligible fashion and written in standard English?

Reviewer #1: Yes

Reviewer #2: No

6. Review Comments to the Author

Reviewer #1: (No Response)

Reviewer #2: Summary: The comments have been incompletely addressed, and there are still numerous spelling/grammatical errors.

Please find below specific comments for each section.

Data availability:

1. Previous comment: The authors state that the data are fully available and included in the manuscript. Is there an appendix with the individual patient data, or where can they be found?

Author reply: only data in table 1, 2, and 3 were collected in this paper. Except the

identifiers (like ID), there is no additional information collected than tables 1, 2, and 3.

If the format of the empty questionnaire is needed, we can upload when requested.

Comment: Data in table 1, 2, and 3 are aggregate data, not individual patient data. It is not possible to replicate the analyses with these data. Can the dataset with the individual patient data be access somewhere? If not, the data availability statement needs to be addressed accordingly.

• As per your good recommendation, the data availability sub section had been added in the manuscript under declaration section, describing all data are included in the manuscript.

Abstract:

1. Previous comment: The abbreviation rrr should be introduced.

Author reply: The abbreviation rrr has been introduced in the “list of abbreviations”

section.

Comment: All abbreviations should be introduced in the manuscript text, not only in the list of abbreviations.

• Response: We accept the comment and rrr is written in expanded form (relative risk ratio) under the title of “data analysis” at last paragraph in its first use. In abstract, since the result section is better to be summary of the findings, we prefer to summarize and to delete the values of regression including the rrr. This could not change the meaning of the results. 

2. Previous comment: In the conclusion, please state the implications of your findings.

Author reply: We accept the comment and correction is made accordingly.

Comment: What changes were made? I don’t see any in the conclusion.

• Response: We accept the comment and implications for the prevalence and risk factors has been specified as seen under conclusion section.

Introduction:

1. Previous comment: Page 3, line 54: Which host factors are the authors referring to?

Author reply: We accept the comment and we refer the participants’ factor as the host

factors.

Comment: It is not enough to refer to participants’ factors instead of host factors. Which participant factors may be relevant?

• Response: We understand your concern and we are referring to the life style (habits) of participants like risk taking behavior, substance abuse, alcohol use e.tc.

Methods:

1. Previous comment: Page 5, line 102: Why did the authors assume a cervical precancer prevalence of 50% for the sample size calculation when a previous institution-based study had found a prevalence of 6.7% in the Tigray region?

Author reply: In Ethiopia, VIA screening was not available for non HIV women until

2015. It was only three hospitals providing VIA screening for HIV patents in the study

area, Tigray. Therefore the above mentioned prevalence is for HIV patients. Moreover,

the above prevalence would provide us less number of participants and it is advisable

to increases sample size if affordable. Therefore, we assume increasing participants

would improve the representativeness of the eligible participants.

Comment: I do not understand this reply. Women living with HIV are at higher risk of developing precancerous cervical lesions. If the estimate of 6.7% stems from a study among women living with HIV, then a lower prevalence should be assumed for the reported study including HIV-negative women.

• Response: Yes, this is logically true. But to initiate our study, we base on a 4 years report from the regional health bureau and PATHFINDER, Ethiopia, Sep, 2014. In the report, 111 HIV unknown status were counseled and all were screened. Based on the result found, 29(26%) were VIA +ve for pre-cancerous cervical lesion with no specified reason. Therefore, we thought that it could be more and taking the 6.7% may provide less number of samples that could not represent the region and we decide to use if there is greater prevalence. We did not find any recent community based study in the region that could give us large sample size; for that reason we consider the prevalence that can provide us maximum sample size (50%). The drawback of this may be maximizing the cost of the study. 

2. Previous comment: Please state which variables were included in the multivariable model finally.

Author reply: Dear/Sir/Madam: the model variables are mentioned under the “factors

associated---,” sub section, in paragraph two.

Response: From that paragraph it is not clear which variables were included in the final multivariable model. Furthermore, the authors should explain how they chose variables for inclusion in the multivariable model. This information should be given in the methods section.

• Response: We apologize for our previous response. As now included under the analysis sub section, the variables with p-value < 0.05 in the unadjusted multinomial regression was considered as significant and used in the multivariable multinomial logistic regression.

3. Eligibility criteria: “all adult women” – please state an age range. Were women eligible for screening from the age of 15 already? What do the national cervical cancer screening guidelines recommend?

• Response: We understand and accept the concern. When we depend on the national guideline, it recommends 30 – 49 years old female to be eligible for screening. This manual is developed based on different developing countries. Meaning, in our set up, there are 9 years old married (starting of sexual contact). The range of 30-49 ages implies that they started sex in the least 5 (immune-compromised) to 10 years (which is at 20 years to 44 years). In our set up, it is different, (which may be 14 - 19 years); for that we can find at least 15 to 19 years old married women stayed 5 to 10 yeast after sexual initiation on the ground. Moreover, this guideline is focusing on the HPV caused cervical cancer screening. But PHV is not the only factor that can cause cervical cancer. As part of our professional activities, we had faced with 16 years old women who had advanced cervical CA. This may be caused by other factors (not HPV) that are not related to early sexual initiation. Therefore, we consider 16 to 65 years old women who are with sexual experience (at least in the last 5 years) provided that they are living independent of their family or with husband for the justification of Ethical clearance. This provides a good coverage of eligible women in the community. 

Results:

1. Previous comment: How was history of STI determined? Symptoms? Treated STI?

Author response: We determine the STI history based on items related to symptom of the

infections taken from the national syndrome approach of the STI guideline.

Comment: This information should be added to the manuscript.

• Response: We accept the comment and we used - having history of STI, if any one of the S/S is present (discharge, offensive secretion, itching, dysuria, lower abdominal pain, and fever). We also added it to the manuscript at the last statement of the data collection procedure sub section. 

2. Previous comment: Why was history of cervical cancer screening not included as a potential predictor? It would have been interesting to see how many women had been screened previously, and whether that was associated with having cervical precancer.

Author response: we respect your valuable comment. First of all, there was no routine

cervical screening mechanism in the country; except for HIV cases started 2011. In

the absence of service, asking utilization is not logical.

Comment: This background information should be added to the manuscript as readers may not be aware of this.

• Response: We accept the concern, and we added it to the manuscript at the last statement of the data collection procedure sub section. 

3. Table 4: For some of the variables (e.g., age) there is no reference category for aRRR. The analysis looks wrong.

• Response: We accept the concern, and 15-30 is the reference and we corrected it in the table (wrongly, last values of the category is pasted to first category in both positive and suspicious outcome variables).

4. Table 4: explanations for *, **, and *** are missing.

• Response: We accept your concern, it is included under the table as NB: P - value < 0.001 = ***, 0.001 - 0. 009 = ** and 0. 010 - 0. 05 = * and cRRR – crude relative risk ratio, aRRR - Adjusted relative risk ratio, C.I. - Confidence Interval

Discussion:

1. Previous comment: Page 12, line 232: HIV status mentioned here, but not in the results section?

Author response: We accept the comment and correction is made accordingly.

Comment: What is the correction? That HIV status is removed entirely? Does that mean that no information was available on HIV status?

• Response: At first, it was included as variable, and the statistician was used the raw data to analyze the draft result. But at that time when we evaluate the data collection process, we addressed that the collected data on the variable ‘HIV’ status was incomplete (some sites has no HIV testing clinic and filled as unknown, which could not be logically true to treat those sites with sites that have HIV clinic). Hence we agreed to discard this variable to prevent error. The problem that we included it at the first draft was because of all Co-authors did not meet in person, since they live at different countries (Addis Ababa, Mekelle - Tigray, German, and Chicago (USA).

2. Previous comment: The authors should discuss the limitations of their study.

Author response: We accept the comment and correction is made accordingly.

Comment: Discussion of study limitations is insufficient.

• Response: We accept the concern, and we have tried to incorporate more limitations as much as we can. We believe limitations beyond the study may not be explanatory for our findings. 

3. Previous comment: The discussion is mainly a comparison with results from other studies. It would be helpful if the authors could also expand on hypotheses and explanations of why certain factors may predict cervical pre-cancer. For example, why would divorced and widowed women be at increased risk of precancer after adjustment for age? Any hypotheses?

Author response: We accept the comment and correction is made accordingly.

Comment: This comment has been inadequately addressed, and the discussion section is difficult to understand sometimes. E.g., “In Ethiopia, long time contraceptive users are few, which is evidenced by having high fertility rate. This may not predispose to cervical pre-cancerious lesion.” – what does that mean? What does not predispose to precancerous cervical lesions? The discussion section needs to be restructured and improved.

• Response: We accept the comment and tried to go through again to incorporate some hypothesis. It can be seen from track changes file.

7. PLOS authors have the option to publish the peer review history of their article (what does this mean?). If published, this will include your full peer review and any attached files.

Do you want your identity to be public for this peer review? For information about this choice, including consent withdrawal, please see our Privacy Policy.

Reviewer #1: No

Reviewer #2: No

---

## [Decision Letter · Decision Letter 2]

6 Sep 2022

PONE-D-20-24267R2Epidemiology of Pre-cancerous Cervical Lesion and Risk Factors among Adult Women n Tigray, EthiopiaPLOS ONE

Dear Dr. Gerezgiher Buruh Abera,

Thank you for submitting your manuscript to PLOS ONE. After careful consideration, we feel that it has merit but does not fully meet PLOS ONE’s publication criteria as it currently stands. Therefore, we invite you to submit a revised version of the manuscript that addresses the points raised during the review process.

We look forward to receiving your revised manuscript.

Kind regards,

Sebsibe Tadesse, PhD

Academic Editor

PLOS ONE

Reviewers' comments:

Reviewer's Responses to Questions

**Comments to the Author**

1. If the authors have adequately addressed your comments raised in a previous round of review and you feel that this manuscript is now acceptable for publication, you may indicate that here to bypass the “Comments to the Author” section, enter your conflict of interest statement in the “Confidential to Editor” section, and submit your "Accept" recommendation.

Reviewer #3: (No Response)

Reviewer #4: (No Response)

2. Is the manuscript technically sound, and do the data support the conclusions?

Reviewer #3: Yes

Reviewer #4: No

3. Has the statistical analysis been performed appropriately and rigorously? 

Reviewer #3: Yes

Reviewer #4: I Don't Know

4. Have the authors made all data underlying the findings in their manuscript fully available?

Reviewer #3: No

Reviewer #4: No

5. Is the manuscript presented in an intelligible fashion and written in standard English?

Reviewer #3: Yes

Reviewer #4: No

6. Review Comments to the Author

Reviewer #3: The most recent version is much better but still needs some improvement. So I have 4 main comments:

1. Yu need to go through the attached document and see all my highlighted arts and suggestions for improvement etc. then do work on them

2. under the methods part what quality assurance measures were put in place apart from the training the nurses had? Did anyone else verify any screening findings?

3. I did not see any write up on what happened to screen positive women. You must always link screening to management so what happened?

4. the limitations part, the key issue is the limitation of VIA for finding lesions, that is important and must be stated. And should should then explain why that was used instead of more sensitive and specific screening modalities

Reviewer #4: 1. Previous comment: The abbreviation rrr should be introduced.

a. Author reply: The abbreviation rrr has been introduced in the “list of abbreviations” section.

b. Comment: All abbreviations should be introduced in the manuscript text, not only in the list of abbreviations.

c. •Response: We accept the comment and rrr is written in expanded form (relative risk ratio) under the title of “data analysis” at last paragraph in its first use. In abstract, since the result section is better to be summary of the findings, we prefer to summarize and to delete the values of regression including the rrr. This could not change the meaning of the results.

Comment on Revision 2: Results section in the abstracts do normally contain the relative risk or adjusted relative risk. I would recommend including the numerical estimates, with the appropriate introduction of the abbreviation in the abstract.

2. Previous comment: In the conclusion, please state the implications of your findings.

a. Author reply: We accept the comment and correction is made accordingly.

b. Comment: What changes were made? I don’t see any in the conclusion.

c. •Response: We accept the comment and implications for the prevalence and risk factors has been specified as seen under conclusion section.

Comment on Revision 2: I do not see any updates in the abstract’s conclusion section on implications of findings.

Introduction:

1. Previous comment: Page 3, line 54: Which host factors are the authors referring to?

a. Author reply: We accept the comment and we refer the participants’ factor as the host factors.

b. Comment: It is not enough to refer to participants’ factors instead of host factors. Which participant factors may be relevant?

c. Response: We understand your concern and we are referring to the life style (habits) of participants like risk taking behavior, substance abuse, alcohol use e.tc. Methods: 1.

Comment on Revision 2: Substance abuse, alcohol use, risk-taking behavior are not risk factors for CC, except as they would work through increase in risk for HPV or HIV. I believe an appropriate way to address this comment would include the specifically identified risk factors which include immune status and smoking.

2. Previous comment: Page 5, line 102: Why did the authors assume a cervical precancer prevalence of 50% for the sample size calculation when a previous institution-based study had found a prevalence of 6.7% in the Tigray region?

a. Author reply: In Ethiopia, VIA screening was not available for non HIV women until 2015. It was only three hospitals providing VIA screening for HIV patents in the study area, Tigray. Therefore the above mentioned prevalence is for HIV patients. Moreover, the above prevalence would provide us less number of participants and it is advisable to increases sample size if affordable. Therefore, we assume increasing participants would improve the representativeness of the eligible participants.

b. Comment: I do not understand this reply. Women living with HIV are at higher risk of developing precancerous cervical lesions. If the estimate of 6.7% stems from a study among women living with HIV, then a lower prevalence should be assumed for the reported study including HIV-negative women.

c. •Response: Yes, this is logically true. But to initiate our study, we base on a 4 years report from the regional health bureau and PATHFINDER, Ethiopia, Sep, 2014. In the report, 111 HIV unknown status were counseled and all were screened. Based on the result found, 29(26%) were VIA +ve for pre-cancerous cervical lesion with no specified reason. Therefore, we thought that it could be more and taking the 6.7% may provide less number of samples that could not represent the region and we decide to use if there is greater prevalence. We did not find any recent community based study in the region that could give us large sample size; for that reason we consider the prevalence that can provide us maximum sample size (50%). The drawback of this may be maximizing the cost of the study.

d. Comment on Revision 2: I am confused by the reply. Overestimating the prevalence of precancer would decrease the overall sample size estimate for the study, making it more feasible, but also more likely to be underpowered to show associations. There are multiple studies from East Africa which show prevalence of VIA positivity, and most are within the rates of 10-30%. As the prior reviewer suggested, using the actual estimate of 6.7% would have facilitated a more appropriate sample size calculation.

3. Previous comment: Please state which variables were included in the multivariable model finally.

a. Author reply: Dear/Sir/Madam: the model variables are mentioned under the “factors associated---,” sub section, in paragraph two.

b. Response: From that paragraph it is not clear which variables were included in the final multivariable model. Furthermore, the authors should explain how they chose variables for inclusion in the multivariable model. This information should be given in the methods section.

c. •Response: We apologize for our previous response. As now included under the analysis sub section, the variables with p-value < 0.05 in the unadjusted multinomial regression was considered as significant and used in the multivariable multinomial logistic regression.

d. Comment on Revision 2: This has been partially addressed. Did the authors use any additional assessments of fit for the final regression model?

4. 3. Eligibility criteria: “all adult women” – please state an age range. Were women eligible for screening from the age of 15 already? What do the national cervical cancer screening guidelines recommend?

a. •Response: We understand and accept the concern. When we depend on the national guideline, it recommends 30 – 49 years old female to be eligible for screening. This manual is developed based on different developing countries. Meaning, in our set up, there are 9 years old married (starting of sexual contact). The range of 30-49 ages that they started sex in the least 5 (immune-compromised) to 10 years (which is at 20 years to 44 years). In our set up, it is different, (which may be 14 - 19 years); for that we can find at least 15 to 19 years old married women stayed 5 to 10 yeast after sexual initiation on the ground. Moreover, this guideline is focusing on the HPV caused cervical cancer screening. But PHV is not the only factor that can cause cervical cancer. As part of our professional activities, we had faced with 16 years old women who had advanced cervical CA. This may be caused by other factors (not HPV) that are not related to early sexual initiation. Therefore, we consider 16 to 65 years old women who are with sexual experience (at least in the last 5 years) provided that they are living independent of their family or with husband for the justification of Ethical clearance. This provides a good coverage of eligible women in the community.

b. Comment on Revision 2: I am concerned that the authors’ state screening women 15-19 or even younger would help identify non-HPV related cervical cancer. These numbers are vanishingly low (<0.01%), and multiple studies across many settings have shown that early screening, even in the setting of early sexual debut will have more harms, and costs, than benefits.

Results:

1. Previous comment: How was history of STI determined? Symptoms? Treated STI?

a. Author response: We determine the STI history based on items related to symptom of the infections taken from the national syndrome approach of the STI guideline. Comment: This information should be added to the manuscript.

b. Response: We accept the comment and we used - having history of STI, if any one of the S/S is present (discharge, offensive secretion, itching, dysuria, lower abdominal pain, and fever). We also added it to the manuscript at the last statement of the data collection procedure sub section.

c. Comment on Revision 2: These was previously resolved

2. Previous comment: Why was history of cervical cancer screening not included as a potential predictor? It would have been interesting to see how many women had been screened previously, and whether that was associated with having cervical precancer.

a. Author response: we respect your valuable comment. First of all, there was no routine cervical screening mechanism in the country; except for HIV cases started 2011. In the absence of service, asking utilization is not logical.

b. Comment: This background information should be added to the manuscript as readers may not be aware of this.

c. •Response: We accept the concern, and we added it to the manuscript at the last statement of the data collection procedure sub section.

d. Comment on Revision 2: This has been addressed, but I also wonder why HIV status was not included as a risk factor?

3. Table 4: For some of the variables (e.g., age) there is no reference category for aRRR. The analysis looks wrong.

a. •Response: We accept the concern, and 15-30 is the reference and we corrected it in the table (wrongly, last values of the category is pasted to first category in both positive and suspicious outcome variables).

b. This was addressed. Table 4 needs additional formatting to ensure visibility of complete variable names.

4. Table 4: explanations for *, **, and *** are missing. •Response: We accept your concern, it is included under the table as NB: P - value < 0.001 = ***, 0.001 - 0. 009 = ** and 0. 010 - 0. 05 = * and cRRR – crude relative risk ratio, aRRR - Adjusted relative risk ratio, C.I. - Confidence Interval

a. Comment on Revision 2: This has been addressed.

Discussion:

1. Previous comment: Page 12, line 232: HIV status mentioned here, but not in the results section?

a. Author response: We accept the comment and correction is made accordingly.

b. Comment: What is the correction? That HIV status is removed entirely? Does that mean that no information was available on HIV status?

c. •Response: At first, it was included as variable, and the statistician was used the raw data to analyze the draft result. But at that time when we evaluate the data collection process, we addressed that the collected data on the variable ‘HIV’ status was incomplete (some sites has no HIV testing clinic and filled as unknown, which could not be logically true to treat those sites with sites that have HIV clinic). Hence we agreed to discard this variable to prevent error. The problem that we included it at the first draft was because of all Co-authors did not meet in person, since they live at different countries (Addis Ababa, Mekelle - Tigray, German, and Chicago (USA).

d. Comment on Revision 2: This absence and reason for the absence of this important variable needs to be addressed,

2. Previous comment: The authors should discuss the limitations of their study.

a. Author response: We accept the comment and correction is made accordingly.

b. Comment: Discussion of study limitations is insufficient.

c. •Response: We accept the concern, and we have tried to incorporate more limitations as much as we can. We believe limitations beyond the study may not be explanatory for our findings.

d. Comment on Revision 2: This is still underdeveloped. This would be a place to include limitations on data for follow-up (VIA is a proxy for precancer and cancer), and lack of data on HIV status.

3. Previous comment: The discussion is mainly a comparison with results from other studies. It would be helpful if the authors could also expand on hypotheses and explanations of why certain factors may predict cervical pre-cancer. For example, why would divorced and widowed women be at increased risk of precancer after adjustment for age? Any hypotheses?

a. Author response: We accept the comment and correction is made accordingly. Comment: This comment has been inadequately addressed, and the discussion section is difficult to understand sometimes. E.g., “In Ethiopia, long time contraceptive users are few, which is evidenced by having high fertility rate. This may not predispose to cervical pre-cancerious lesion.” – what does that mean? What does not predispose to precancerous cervical lesions? The discussion section needs to be restructured and improved.

b. •Response: We accept the comment and tried to go through again to incorporate some hypothesis. It can be seen from track changes file.

c. Comment on Revision 2: I feel like this comment has still not been appropriately addressed. The discussion still reads as a list of comparisons, rather than an exploration of the reasons for these differences. Major restructuring would improve this section. The conclusions sections does not make sense either—cervical cancer prevention through “preventable risk factors” is not as effective as through HPV testing and vaccination. I don’t think study of the age distribution of pre-cancerous lesions makes as much sense as study of the relationship between VIA and diagnosis of pre-cancer or cancer.

7. PLOS authors have the option to publish the peer review history of their article (what does this mean?). If published, this will include your full peer review and any attached files.

Reviewer #3: No

Reviewer #4: No

---

## [Author Response · Author response to Decision Letter 2]

7 Dec 2022

Date: September 14_2022

To: PLOS ONE" plosone@plos.org

From: Gerezgiher Buruh Abera" gbamsc2002@gmail.com (PI)

Subject: Point by Point response to editor and reviewer(s).

Title Epidemiology of Cervical Precancerous Lesion and Risk Factors Among Adult Women In Tigray, Ethiopia

Reference PONE-D-20-24267R2

 Dear/Sir/ professors: editor and reviewers:

Thank you for your constructive and supportive comments. We are very grateful for the consideration of the manuscript. In accordance with the editors and reviewers’ valuable comments and recommendations, we have revised the manuscript and we are hereby submitting the revised work for your consideration.

Reviewers' comments:

Reviewer's Responses to Questions

1. If the authors have adequately addressed your comments raised in a previous round of review and you feel that this manuscript is now acceptable for publication, you may indicate that here to bypass the “Comments to the Author” section, enter your conflict of interest statement in the “Confidential to Editor” section, and submit your "Accept" recommendation.

Reviewer #3: (No Response)

Reviewer #4: (No Response)

2. Is the manuscript technically sounds, and do the data support the conclusions?

Reviewer #3: Yes

Reviewer #4: No

3. Has the statistical analysis been performed appropriately and rigorously? 

Reviewer #3: Yes

Reviewer #4: I Don't Know

4. Have the authors made all data underlying the findings in their manuscript fully available?

Reviewer #3: No

Reviewer #4: No

5. Is the manuscript presented in an intelligible fashion and written in Standard English?

Reviewer #3: Yes

Reviewer #4: No

6. Review Comments to the Author

Reviewer #3: 

The most recent version is much better but still needs some improvement. So I have 4 main comments:

1. Yu need to go through the attached document and see all my highlighted arts and suggestions for improvement etc. then do work on them.

• Response: Thank you for detail comments on the attachment. We went through the attached pdf and tried to address comments highlighted. We accept all comments and that help us to made more correction in the manuscript.

2. Under the methods part what quality assurance measures were put in place apart from the training the nurses had? Did anyone else verify any screening findings?

• Response: We accept the comment and the provider was supported by the guideline protocol of VIA screening and supervised by gynecologists and assigned supervisors available in the institutions as appropriate to assure the quality of data collection. We include the quality steps in the main text as well.

3. I did not see any write up on what happened to screen positive women. You must always link screening to management so what happened?

• Response: We accept your comment and each client with positive or suspicious result was liked to related medical centers for further investigation and/or management. Most data collection sites have at least Cryotherapy for management and Gynecologic algorithm was used to follow clients with certain sign of pre-cancerous lesion. This is also added in the main text under the “Risk factors of cervical cancer:” subsection at page 11.

4. The limitations part, the key issue is the limitation of VIA for finding lesions, that is important and must be stated. And should then explain why that was used instead of more sensitive and specific screening modalities

• Response: We accept your comment and we added that other cervical screening are better identify pre-cancerous lesion, though the methods are not available in the study sites and are expensive than VIA. 

Reviewer #4: 

1. Previous comment: The abbreviation rrr should be introduced.

a. Author reply: The abbreviation rrr has been introduced in the “list of abbreviations” section.

b. Comment: All abbreviations should be introduced in the manuscript text, not only in the list of abbreviations.

c. •Response: We accept the comment and rrr is written in expanded form (relative risk ratio) under the title of “data analysis” at last paragraph in its first use. In abstract, since the result section is better to be summary of the findings, we prefer to summarize and to delete the values of regression including the rrr. This could not change the meaning of the results.

Comment on Revision 2: Results sections in the abstracts do normally contain the relative risk or adjusted relative risk. I would recommend including the numerical estimates, with the appropriate introduction of the abbreviation in the abstract.

• Response: We accept the comment and effects size estimations are included in the abstract with introduction of the rrr. We introduced the rrr in the abstract, under subsection “Factors associated with pre-cancerous Cervical Lesion) of page 12 and as footnote of table 4.

2. Previous comment: In the conclusion, please state the implications of your findings.

a. Author reply: We accept the comment and correction is made accordingly.

b. Comment: What changes were made? I don’t see any in the conclusion.

c. •Response: We accept the comment and implications for the prevalence and risk factors have been specified as seen under conclusion section.

Comment on Revision 2: I do not see any updates in the abstract’s conclusion section on implications of findings.

• Response: We accept the comment and sorry for that we missed correcting in the abstract, now we added the statement as “The prevalence of pre-cancerous cervical lesion is high as compared to other regional prevalence in the country. This finding implies that the sexual exposure, having no permanent husband and being not educated attributes to the high prevalence of pre-cancerous cervical lesion and may aggravate the transmission of HPV.”

Introduction:

1. Previous comment: Page 3, line 54: Which host factors are the authors referring to?

a. Author reply: We accept the comment and we refer the participants’ factor as the host factors.

b. Comment: It is not enough to refer to participants’ factors instead of host factors. Which participant factors may be relevant?

c. Response: We understand your concern and we are referring to the life style (habits) of participants like risk taking behavior, substance abuse, alcohol use e.tc. Methods: 1.

Comment on Revision 2: Substance abuse, alcohol use, risk-taking behavior are not risk factors for CC, except as they would work through increase in risk for HPV or HIV. I believe an appropriate way to address this comment would include the specifically identified risk factors which include immune status and smoking.

• Response: Dear/Sir, sorry for confusing you about our response statements, as per the Ethiopian cervical cancer prevention and control guideline, we tried to see if similar risk factors found in different researches. The ‘host factor” is mentioned in other studies as reference 5 and 6, but it is not ass variable of our study. There are many associated factors in different studies, example: risk taking behavior like sex without condom use, substance use including alcohol use or smoking and exposing to STI as risk factors for cervical cancer. Therefore we are referring to those factors as participant’s risk factors to unsafe sex and exposing to HPV, as HPV is prevalent STI. HPV is most common cause of cervical cancer. Hence those participants’ factors could have association with cervical cancer.

2. Previous comment: Page 5, line 102: Why did the authors assume a cervical pre-cancer prevalence of 50% for the sample size calculation when a previous institution-based study had found a prevalence of 6.7% in the Tigray region?

a. Author reply: In Ethiopia, VIA screening was not available for non HIV women until 2015. It was only three hospitals providing VIA screening for HIV patents in the study area previously, Tigray. Therefore the above mentioned prevalence is for HIV patients. Moreover, the above prevalence would provide us less number of participants and it is advisable to increases sample size if affordable. Therefore, we assume increasing participants would improve the representativeness of the eligible participants.

b. Comment: I do not understand this reply. Women living with HIV are at higher risk of developing precancerous cervical lesions. If the estimate of 6.7% stems from a study among women living with HIV, then a lower prevalence should be assumed for the reported study including HIV-negative women.

c. •Response: Yes, this is logically true. But to initiate our study, we base on a 4 years report from the regional health bureau and PATHFINDER Ethiopia, Sep, 2014. In the report, 111 HIV unknown status were counseled and all were screened. Based on the result found, 29(26%) were VIA +ve for pre-cancerous cervical lesion with no specified reason. Therefore, we thought that it could be more and taking the 6.7% may provide less number of samples that could not represent the region and we decide to use if there is greater prevalence. We did not find any recent community based study in the region that could give us large sample size; for that reason we consider the prevalence that can provide us maximum sample size (50%). The drawback of this may be maximizing the cost of the study.

d. Comment on Revision 2: I am confused by the reply. Overestimating the prevalence of precancer would decrease the overall sample size estimate for the study, making it more feasible, but also more likely to be underpowered to show associations. There are multiple studies from East Africa which show prevalence of VIA positivity, and most are within the rates of 10-30%. As the prior reviewer suggested, using the actual estimate of 6.7% would have facilitated a more appropriate sample size calculation.

• Response: We accept that it was good if the prevalence of cervical cancer among the HIV cases based screening to be used to calculate the sample size. The contextual differences between countries make us not to use other countries prevalence. We were standing on the fact that it could be good if all the communities included in the study, hence increasing participants would increase the power of the study. Because a prevalence cold give us minimum possible sample size, provided that maximizing is promoted. Besides, in the absence of prevalence among HIV unknown stats, in the study area, 50% is an alternative. 

Three options are commonly used to determine sample size; the survey based, pre-existing study, and 50%. Therefore, at the conception of this project, we agreed to use 50%, as other alternative was not convenient. Being at this stage of the manuscript, we are happy if we get recommendation on the sample determination modification, but we believe it is difficult to modify the proportion size at this stage. 

3. Previous comment: Please state which variables were included in the multivariable model finally.

a. Author reply: Dear/Sir/Madam: the model variables are mentioned under the “factors associated,” sub section, in paragraph two.

b. Response: From that paragraph it is not clear which variables were included in the final multivariable model. Furthermore, the authors should explain how they chose variables for inclusion in the multivariable model. This information should be given in the methods section.

c. •Response: We apologize for our previous response. As now included under the analysis sub section, the variables with p-value < 0.05 in the unadjusted multinomial regression was considered as significant and used in the multivariable multinomial logistic regression.

d. Comment on Revision 2: This has been partially addressed. Did the authors use any additional assessments of fit for the final regression model?

• Response: Variables like age, marital and education status, occupation, symptoms of cervical cancer, parity, STI history of partner, and lower abdominal pain were included in the multivariate analysis. Those variables were selected based on the p-value < 0.05 in the unadjusted multinomial regression. No other tests were used to select the model variables.

4. 3. Eligibility criteria: “all adult women” – please state an age range. Were women eligible for screening from the age of 15 already? What do the national cervical cancer screening guidelines recommend?

a. • Response: We understand and accept the concern. When we depend on the national guideline, it recommends 30 – 49 years old female to be eligible for screening. This manual is developed based on different developing countries. Meaning, in our set up, there are 9 years old married (starting of sexual contact). The range of 30-49 ages that they started sex in the least 5 (immune-compromised) to 10 years (which is at 20 years to 44 years). In our set up, it is different, (which may be 14 - 19 years); for that we can find at least 15 to 19 years old married women stayed 5 to 10 yeast after sexual initiation on the ground. Moreover, this guideline is focusing on the HPV caused cervical cancer screening. But PHV is not the only factor that can cause cervical cancer. As part of our professional activities, we had faced with 16 years old women who had advanced cervical CA. This may be caused by other factors (not HPV) that are not related to early sexual initiation. Therefore, we consider 16 to 65 years old women who are with sexual experience (at least in the last 5 years) provided that they are living independent of their family or with husband for the justification of Ethical clearance. This provides a good coverage of eligible women in the community.

b. Comment on Revision 2: I am concerned that the authors’ state screening women 15-19 or even younger would help identify non-HPV related cervical cancer. These numbers are vanishingly low (<0.01%), and multiple studies across many settings have shown that early screening, even in the setting of early sexual debut will have more harms, and costs, than benefits.

• Response: We accept your concern, but as per our senior gynecologists (member of the research) recommendation, we based on the community marital status and accepted as participants if they are married and had at least child, being in the age of 16 and above. 

Results:

1. Previous comment: How was history of STI determined? Symptoms? Treated STI?

a. Author response: We determine the STI history based on items related to symptom of the infections taken from the national syndrome approach of the STI guideline. 

Comment: This information should be added to the manuscript.

b. Response: We accept the comment and we used - having history of STI, if any one of the S/S is present (discharge, offensive secretion, itching, dysuria, lower abdominal pain, and fever). We also added it to the manuscript at the last statement of the data collection procedure sub section.

c. Comment on Revision 2: These was previously resolved

• Response: Thank you for helping us to address the above mentioned statement in the manuscript. 

2. Previous comment: Why was history of cervical cancer screening not included as a potential predictor? It would have been interesting to see how many women had been screened previously, and whether that was associated with having cervical precancer.

a. Author response: we respect your valuable comment. First of all, there was no routine cervical screening mechanism in the country; except for HIV cases started 2011. In the absence of service, asking utilization is not logical.

b. Comment: This background information should be added to the manuscript as readers may not be aware of this.

c. •Response: We accept the concern, and we added it to the manuscript at the last statement of the data collection procedure sub section.

d. Comment on Revision 2: This has been addressed, but I also wonder why HIV status was not included as a risk factor?

• We accept the comment and the variable HIV status was included in our questionnaire, but excluded from analysis, because it was filled incomplete, since some sites has no HIV testing clinic and filled as unknown, which could not be logically true to treat those sites with sites that have HIV clinic. This has been added in the text at page 11 above table 2.

3. Table 4: For some of the variables (e.g., age) there is no reference category for aRRR. The analysis looks wrong.

a. •Response: We accept the concern, and 15-30 is the reference and we corrected it in the table (wrongly, last values of the category is pasted to first category in both positive and suspicious outcome variables).

b. This was addressed. Table 4 needs additional formatting to ensure visibility of complete variable names.

• Thank you for making us to see the table again, we tried to make the values visible by stretching the table space especially in the clean manuscript.

4. Table 4: explanations for *, **, and *** are missing. 

•Response: We accept your concern, it is included under the table as NB: P - value < 0.001 = ***, 0.001 - 0. 009 = ** and 0. 010 - 0. 05 = * and cRRR – crude relative risk ratio, aRRR - Adjusted relative risk ratio, C.I. - Confidence Interval

a. Comment on Revision 2: This has been addressed.

• Thank you for confirmation of the response, your comment makes us to see our manuscript in different view.

Discussion:

1. Previous comment: Page 12, line 232: HIV status mentioned here, but not in the results section?

a. Author response: We accept the comment and correction is made accordingly.

b. Comment: What is the correction? That HIV status is removed entirely? Does that mean that no information was available on HIV status?

c. •Response: At first, it was included as variable, and the statistician was used the raw data to analyze the draft result. But at that time when we evaluate the data collection process, we addressed that the collected data on the variable ‘HIV’ status was incomplete (some sites has no HIV testing clinic and filled as unknown, which could not be logically true to treat those sites with sites that have HIV clinic). Hence we agreed to discard this variable to prevent error. The problem that we included it at the first draft was because of all Co-authors did not meet in person, since they live at different countries (Addis Ababa, Mekelle - Tigray, German, and Chicago (USA).

d. Comment on Revision 2: This absence and reason for the absence of this important variable needs to be addressed,

• Response: We accept the comment and we added the reason of removal of the “HIV status” variable under “risk factors of cervical cancer” subsection at last paragraph of page 11 above table 2, as follows “the variable HIV status was excluded from analysis, since it was filled incomplete, because some sites has no HIV testing clinic”, which is illogical to treat sites that has no HIV testing with those that has testing centers.

2. Previous comment: The authors should discuss the limitations of their study.

a. Author response: We accept the comment and correction is made accordingly.

b. Comment: Discussion of study limitations is insufficient.

c. •Response: We accept the concern, and we have tried to incorporate more limitations as much as we can. We believe limitations beyond the study may not be explanatory for our findings.

d. Comment on Revision 2: This is still underdeveloped. This would be a place to include limitations on data for follow-up (VIA is a proxy for pre-cancer and cancer), and lack of data on HIV status.

• Response: Thank you for helping us to see the limitation again. We tried to add the missed variables, used design and screening method other than VIA as limitation in the main text.

3. Previous comment: The discussion is mainly a comparison with results from other studies. It would be helpful if the authors could also expand on hypotheses and explanations of why certain factors may predict cervical pre-cancer. For example, why would divorced and widowed women be at increased risk of precancer after adjustment for age? Any hypotheses?

a. Author response: We accept the comment and correction is made accordingly. 

Comment: This comment has been inadequately addressed, and the discussion section is difficult to understand sometimes. E.g., “In Ethiopia, long time contraceptive users are few, which is evidenced by having high fertility rate. This may not predispose to cervical pre-cancerious lesion.” – What does that mean? What does not predispose to precancerous cervical lesions? The discussion section needs to be restructured and improved.

b. •Response: We accept the comment and tried to go through again to incorporate some hypothesis. It can be seen from track changes file.

c. Comment on Revision 2: I feel like this comment has still not been appropriately addressed. The discussion still reads as a list of comparisons, rather than an exploration of the reasons for these differences. Major restructuring would improve this section. The conclusions sections does not make sense either—cervical cancer prevention through “preventable risk factors” is not as effective as through HPV testing and vaccination. I don’t think study of the age distribution of pre-cancerous lesions makes as much sense as study of the relationship between VIA and diagnosis of pre-cancer or cancer.

• We accept and agreed with your comments. We tried still to synthesis and justify the findings in the discussion part as can be seen from the track change of the manuscript. Or limitation is that we have no internet right now to find backgrounds of cited countries for justification and comparison of the other studies. It is a fact that HPV testing and vaccination is effective to prevent cervical cancer than doing on risk factors, but, it is not available in our setup and dealing with the absence of the service may not give meaning. The preventable risk factors could be in the capacity of poor countries like our country Ethiopia. In research findings, contraceptive use is mentioned as associated factors for cervical cancer. This could be because of the long term use of contraceptive example pills, expose them to STI, including HPV. This has been added to the main text

7. PLOS authors have the option to publish the peer review history of their article (what does this mean?). If published, this will include your full peer review and any attached files.

Do you want your identity to be public for this peer review? For information about this choice, including consent withdrawal, please see our Privacy Policy.

Reviewer #3: No

Reviewer #4: No

---

## [Editor Report · Decision Letter 3]

22 Dec 2022

Epidemiology of Pre-cancerous Cervical Lesion and Risk Factors among Adult Women in Tigray, Ethiopia

PONE-D-20-24267R3

Dear Dr. Gerezgiher Buruh Abera,

We’re pleased to inform you that your manuscript has been judged scientifically suitable for publication and will be formally accepted for publication once it meets all outstanding technical requirements.

Kind regards,

Sebsibe Tadesse, PhD

Academic Editor

PLOS ONE

---

## [Editor Report · Acceptance letter]

29 Dec 2022

PONE-D-20-24267R3 

Epidemiology of Pre-cancerous Cervical Lesion and Risk Factors among Adult Women n Tigray, Ethiopia 

Dear Dr. Abera:

I'm pleased to inform you that your manuscript has been deemed suitable for publication in PLOS ONE. Congratulations! Your manuscript is now with our production department. 

Kind regards, 

on behalf of

Dr. Sebsibe Tadesse 

Academic Editor

PLOS ONE